# Effect of aging on the human myometrium at single-cell resolution

Paula Punzon-Jimenez [1,2,3,11], Alba Machado-Lopez [1,2,11], Raul Perez-Moraga [1,4], Jaime Llera-Oyola [1], Daniela Grases[5], Marta Galvez-Viedma[1], Mustafa Sibai[5], Elena Satorres-Perez [6], Susana Lopez-Agullo[6], Rafael Badenes [7,8], Carolina Ferrer-Gomez[9], Eduard Porta-Pardo[5], Beatriz Roson [1,2], Carlos Simon [1,2,3,10] ✉ & Aymara Mas [1,2] ✉

Age-associated myometrial dysfunction can prompt complications during pregnancy and labor, which is one of the factors contributing to the 7.8-fold increase in maternal mortality in women over 40. Using single-cell/single-nucleus RNA sequencing and spatial transcriptomics, we have constructed a cellular atlas of the aging myometrium from 186,120 cells across twenty perimenopausal and postmenopausal women. We identify 23 myometrial cell subpopulations, including contractile and venous capillary cells as well as immune-modulated fibroblasts. Myometrial aging leads to fewer contractile capillary cells, a reduced level of ion channel expression in smooth muscle cells, and impaired gene expression in endothelial, smooth muscle, fibroblast, perivascular, and immune cells. We observe altered myometrial cell-to-cell communication as an aging hallmark, which associated with the loss of 25 signaling pathways, including those related to angiogenesis, tissue repair, contractility, immunity, and nervous system regulation. These insights may contribute to a better understanding of the complications faced by older individuals during pregnancy and labor.

The upward trend in human longevity represents a challenge to healthcare systems worldwide, especially when considering women's health[1]. According to the World Health Organization (https://www.who.int/news-room/fact-sheets/detail/ageing-and-health), 1 in 6 of the world's population will be 60 or over by 2030[2].

Aging impairs reproduction, pregnancy, and parturition[3]. Menopause, which typically occurs between 45 and 55, is caused by irreversible ovarian demise that prompts the cessation of menses. Despite being previously associated with the end of a woman's reproductive lifespan[4], an increased accessibility to assisted reproductive techniques has bypassed infertility-associated difficulties related to advanced maternal age; however, the mortality associated with pregnancy and labor complications in older individuals remains problematic. Tangible proof comes from the 7.8-fold increase in maternal mortality in women aged >40 (107.9 deaths per 100,000 live births) compared to those under 25[5,6], which may also be increased by other age-associated conditions such as hypertension or diabetes[7].

[1]Carlos Simon Foundation, Valencia, Spain. [2]Instituto de Investigación Sanitaria INCLIVA, Valencia, Spain. [3]Department of Pediatrics, Obstetrics and Gynecology, University of Valencia, Valencia, Spain. [4]R&D Department, Igenomix, Valencia, Spain. [5]Josep Carreras Leukaemia Research Institute (IJC), Barcelona, Spain. [6]Hospital Universitario y Politécnico La Fe, Valencia, Spain. [7]Department of Surgery, University of Valencia, Valencia, Spain. [8]Hospital Clinico Universitario, Valencia, Spain. [9]Hospital General Universitario de Valencia, Valencia, Spain. [10]Department of Obstetrics and Gynecology, BIDMC, Harvard University, Boston, MA, USA. [11]These authors contributed equally: Paula Punzon-Jimenez, Alba Machado-Lopez. ✉e-mail: carlos.simon@uv.es; amas@fundacioncarlossimon.com

The uterus is a contractile organ essential for reproduction, pregnancy, and labor. Several studies have linked myometrial dysfunction to embryo implantation failure, apart from the rise in worse obstetric outcomes such as preterm birth, labor dystocia, and uterine atony, along with cesarean section or instrumental vaginal delivery which directly impact women's health[8–11]. The thinning of the myometrium during menopause occurs alongside vasculature fibrosis and reduced contractility[3]; however, we lack in-depth knowledge regarding the effects of aging on the myometrium regarding cellular and tissue structural changes. Advancements in single-cell/single-nucleus RNA sequencing (scRNA-seq/ snRNA-seq)-based analyses have provided a deeper insight into the human uterus[12] and the human myometrium in the context of parturition[13], and fibroids[14]. Nonetheless, the lack of studies investigating the human myometrium in the context of aging is our motivation to undertake this research.

We performed scRNA-seq/snRNA-seq and spatial transcriptomics in the human myometrium during menopause to provide a comprehensive description of the molecular and cellular fingerprint associated with age-related myometrial dysfunction. We now report significant age-related alterations to cell type abundance, gene expression patterns in specific cell types, and integrated cell-to-cell communication (CCC) networks.

## Results

### Single-cell atlas of the aging myometrium

To assess the cellular-level effects of aging on the myometrium, we examined myometrial tissue from the fundus and anterior/posterior walls of the uterus of 20 patients – 6 from perimenopausal women (≤54 years old) and 14 from postmenopausal women (>54 years old) (Supplementary Fig. 1A).

After quality control and batch correction (Supplementary Fig. 1B), our dataset comprised scRNA-seq data from 161,202 cells and snRNA-seq data from 24,918 nuclei, which provided an integrated single-cell cellular atlas capturing the cellular landscape of the human menopausal myometrium integrating data from all uterine zones and menopausal states (Supplementary Fig. 1C, D). The main clinical variables (site of collection, vital status, parity or live births and c-section) that might affect the expression patterns in any cell population were also considered for further analysis (Supplementary Data 1, Supplementary Fig. 2).

Cell clustering represented in the uniform manifold approximation and projection (UMAP) space and manual curation of canonical markers[15] identified five major cell types: endothelial cells, fibroblasts, smooth muscle cells (SMCs), perivascular (PV) cells and immune cells (split into myeloid, lymphoid and mast cells). The remaining clusters represented cells of the epithelium, lymphatic endothelium (LEC), and peripheral nervous system (PNS) (Fig. 1A, Supplementary Data 2).

The endothelial cell population expressed *FLT1, CCL14, IFI27, PLEKHG1, LIFR* (Fig. 1B) and canonical genes such as *VWF* and *ENG* (Supplementary Fig. 3A), which are involved in the maintenance of hemostasis through vasculogenesis, blood vessel morphogenesis, and angiogenesis. The fibroblast population specifically expressed *LAMA2, DCN, C7, APOD, NEGR1* (Fig. 1B), and *LUM* (Supplementary Fig. 3A), which function in collagen organization, immune response, and cell adhesion. The SMC population overexpressed *DDP6, FRMD6-AS2, KCNMA1, LINGO2,* and *PDLIM3* (Fig. 1B), which participate, together with canonical genes such as *ACTA2* and *DES* (Supplementary Fig. 3A), in muscle structure development and contractility-associated functions. PV cells overexpressed *RGS6, c11orf96, ADGRL3, RYR2,* and *MT1A* (Fig. 1B), which, alongside *MYH11* and *ACTA2* (Supplementary Fig. 3A), regulate G protein-coupled receptor signaling cascades and calcium channel activity. The immune population overexpressed canonical markers such as *CD3E, TRBC2, AIF1, CYBB, TPSB2,* or *TPSAB1* (Fig. 1B, Supplementary Fig. 3A).

We next systematically mapped the spatial distribution of main cell populations using sc/snRNA-seq within the myometrial tissue architecture (Fig. 1C). We observed a wide distribution of fibroblasts and SMCs, while endothelial and PV cells remained confined to blood vessel-associated regions. The presence of a small subset of epithelial cells correlated with remaining endometrium by histological examination (Supplementary Fig. 3B), that was excluded from further analyses. Lastly, given the critical role of CCC in tissue homeostasis, we examined potential age-related disruption by inferring intercellular communication by analyzing the expression of ligand-receptor pairs using CellChat[16]. Interestingly, CCC network analysis revealed significant changes in information flow (i.e. the total sum of communication probability in the inferred network) for a total of 95 pathways, and the complete loss of signals related to contractility, angiogenesis, tissue repair, nervous system regulation, and anti-inflammatory processes in the postmenopausal myometrium (Fig. 1D). Next, we explored the impact of aging on individual myometrial cell types. Focusing on each major cell type, we delineated cell subtypes by curating gene markers and conducted comparative analyses in the perimenopausal and postmenopausal myometria as our model of aging to decipher abundance alterations, gene expression, CCC, and spatial distribution of cell subtypes.

### Endothelial dysfunction associated with aging

We assessed the impact of aging on myometrial endothelial cells by analyzing four main subpopulations – arterial, contractile capillary, venous capillary, and venous cells (Fig. 2A, Supplementary Data 3). Differential abundance analysis (Fig. 2B) found significant changes associated with myometrial aging, revealing a lower abundance of contractile capillary cells postmenopause (Fig. 2C). Differential gene expression analysis revealed deregulated profiles in postmenopausal vs. perimenopausal myometria (Supplementary Data 4); specifically, arterial cells overexpressed *CCN1*, venous, venous capillary, and contractile capillary cells overexpressed *NFATC2*, while arterial, venous, and venous capillary cells overexpressed *ANO2* in postmenopausal myometria (Fig. 2D). We also observed downregulated *TAGLN* in venous and venous capillary cells, *MYL9* and *MYH11* in venous capillary cells, and *ACTG2* and *FLNA* in contractile capillaries compared to perimenopausal myometria (Fig. 2D). Spatial transcriptomics confirmed a significant reduction in contractile capillary cells within the postmenopausal compared to the perimenopausal myometrium (Fig. 2E, F).

These findings suggest that aging occurs alongside an impairment in the myometrial endothelial cell population supported by reduced cell numbers and contractility-associated gene expression.

### Transcriptomic changes in aging myometrial fibroblasts

We identified eight distinct myometrial fibroblast subpopulations based on transcriptomic profiles (Fig. 3A, B, Supplementary Data 5). We found that classic fibroblasts expressed *RARRES2, SELENOP, RPS10, MPG,* or *APOD*, while the expression of *CCL2, GPRC5A, CYTOR, or IL1R1* characterized immune-modulated fibroblasts. Intermediate fibroblasts specifically expressed *NRP1*, while myofibroblasts expressed *MYH11, DPP6,* and *KNMA1*. Nervous system regulatory fibroblasts expressed genes associated with nervous system regulation (e.g., *NLGN1* or *NRP2*), and stressed fibroblasts expressed genes involved in stress response and DNA damage. Additionally, we labeled two further subclusters of "universal" fibroblasts based on *COL15A1* and *PI16* expression[17].

While we did not observe any age-related alterations to cell abundance in the fibroblastic compartment during myometrial aging, differential expression analysis revealed an increase in aging-associated (e.g., *FOS* and *FOSB*) and immune system regulation (e.g., *NFATC2*) gene expression in universal PI16+ and COL15A1+, classic, and intermediate NRP1+ fibroblasts in the postmenopausal myometria

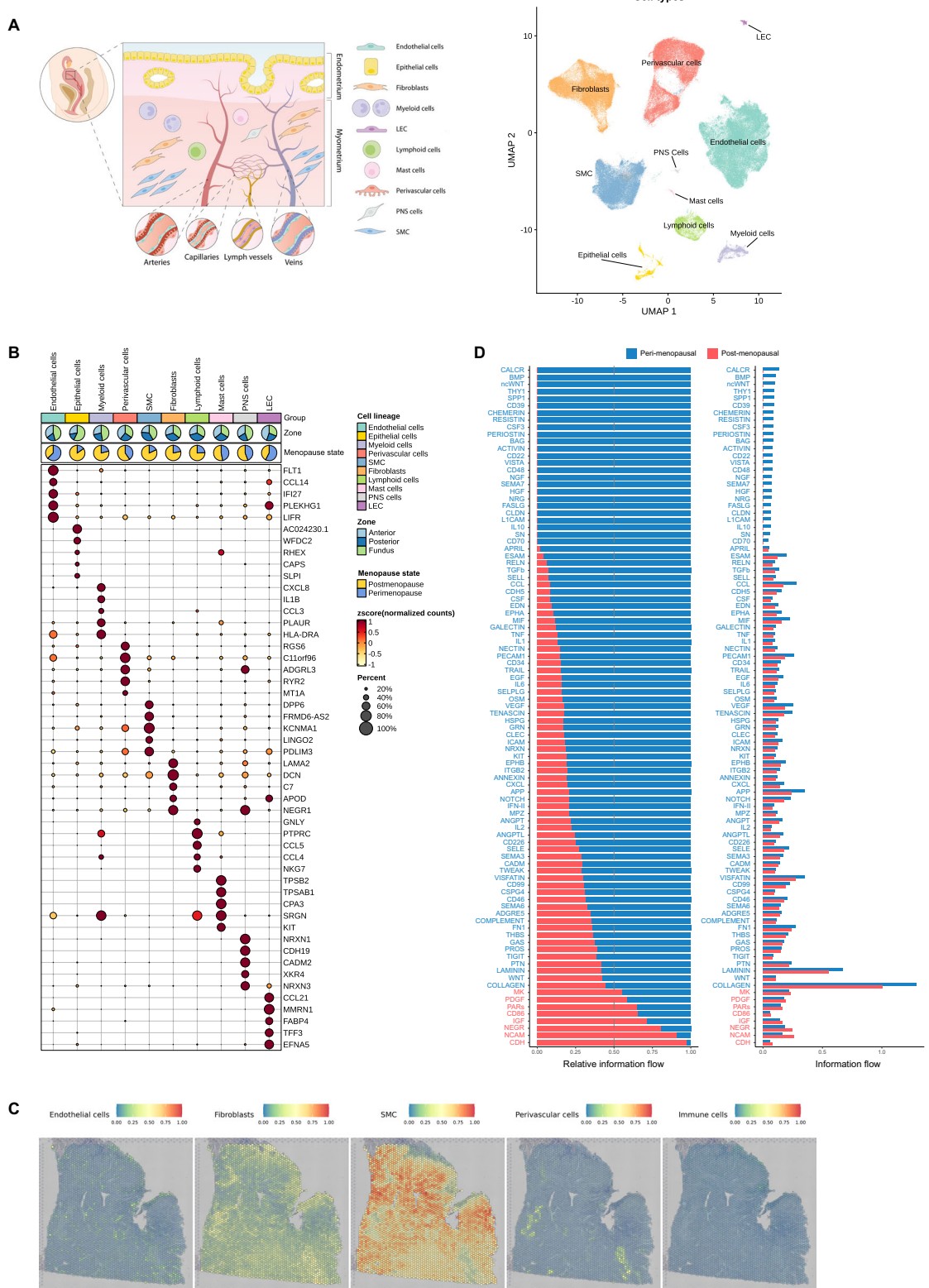

**Fig. 1 | Integrated Single-Cell Atlas of the Human Myometrium. A** Schematic representation of the human uterus and identified cell types within the myometrium (left). Visualization of uniform manifold approximation and projection (UMAP) showing an integrative clustering of high-quality cells/nuclei from human myometria (perimenopausal, *n* = 6; postmenopausal, *n* = 14) at 0.5 resolution (right). **B** Dotplot indicating the relative expression of each identified cell type's top five discriminatory genes (color indicates average expression, while dot size represents the percentage of cells expressing specific genes). Pie charts illustrate the proportion of cells according to menopausal state (perimenopause in blue, postmenopause in yellow) and myometrial zone (light blue -anterior region, dark blue - posterior region, and green - fundus). **C** Panels displaying the spatial location of endothelial cells, fibroblasts, SMCs, PV cells, and immune cells in the myometrium. **D** Relative and absolute flows of differentially active signaling pathways during myometrial aging (comparing perimenopausal and postmenopausal myometria). LECs lymphatic endothelial cells, PNS peripheral nervous system, SMCs smooth muscle cells.

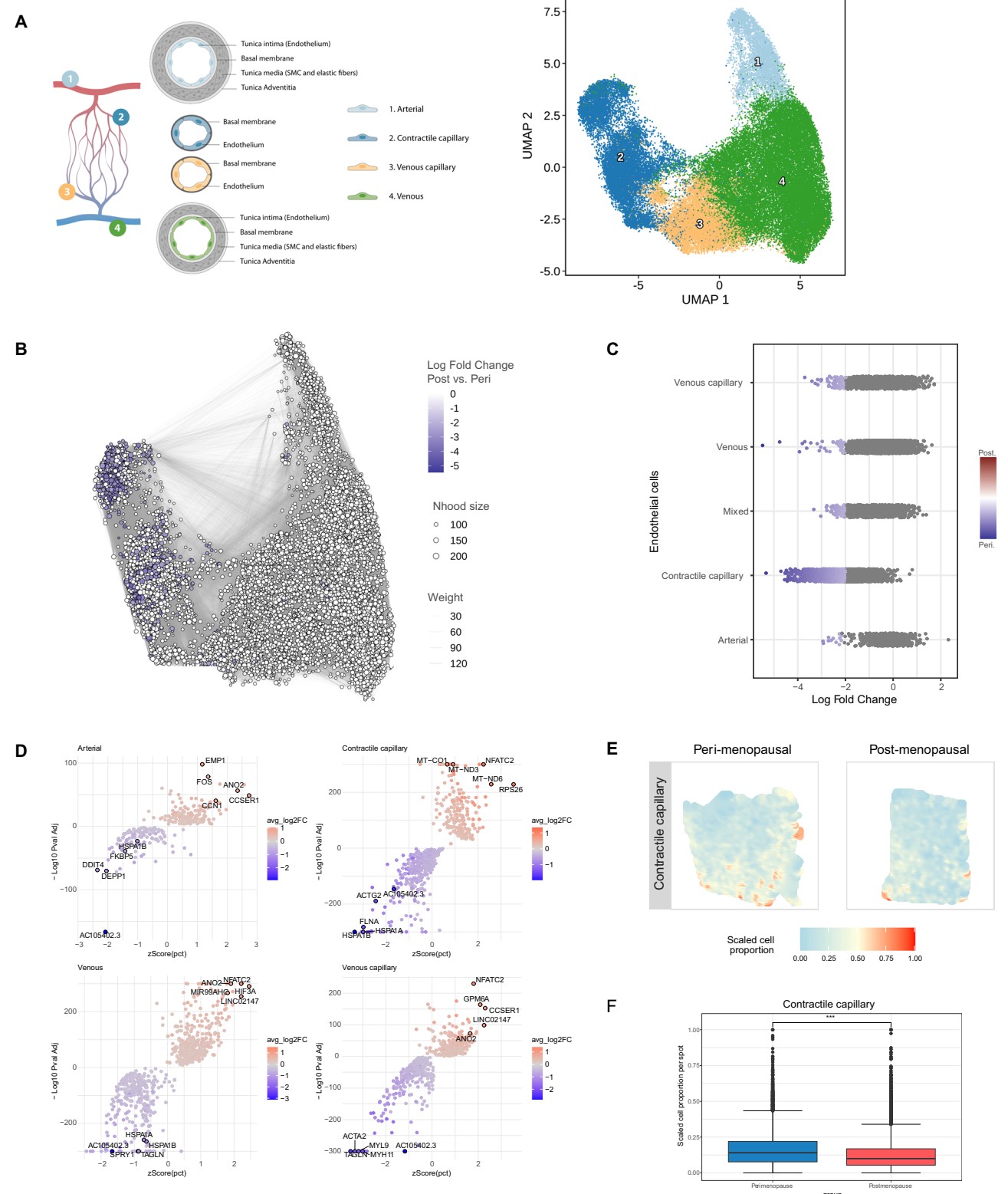

(Fig. 3C; Supplementary Data 6). During myometrial aging, fibroblasts (excluding nervous system regulatory fibroblasts) exhibited reduced expression of genes involved in collagen homeostasis (e.g., *APOD, COL1A1, COL1A2,* and *COL3A1*) (Fig. 3C, Supplementary Data 6). *PABPC1* was the only differentially expressed gene encountered in the nervous system regulatory fibroblasts (Supplementary Data 6). Spatial transcriptomics also confirmed the increased expression of collagen (*COL5A3*) and senescence (*CCN1*) markers during aging (Fig. 3D, E).

Overall, our transcriptomic analysis suggested significant changes in myofibroblast, universal (*COL15A1, PI16*), classic, immune-modulated, stressed, and NRP1+ intermediate fibroblasts during myometrial aging.

**Less contractile and ion-conductive smooth muscle cells**

We identified four SMC subtypes - canonical, contractile, stimuli-response, and stressed – in the myometrium (Fig. 4A). The canonical

**Fig. 2 | Endothelial Dysfunction in the Aging Myometrium. A** Schematic representation of the endothelial cells (left) and UMAP visualization of the principle endothelial subpopulations in the myometrium at 0.5 resolution (right). **B** Neighborhood graph highlighting the differential abundance of endothelial cells in the aging myometrium. Dot size represents neighborhoods, while edges depict the number of cells shared between neighborhoods. Neighborhoods colored in blue represent those with a significant decrease in cell abundance during myometrial aging. **C** Beeswarm plot of differential cell abundance by cell type. X-axis represents the log-fold change in abundance during myometrial aging. Each dot represents a neighborhood; neighborhoods colored in blue represent those with a significant decrease in cell abundance in postmenopausal myometria. **D** Volcano plots representing age-related differentially expressed genes in each endothelial cell subpopulation. Positive LogFC indicates overexpression in the postmenopause myometrium, whereas negative LogFC indicates overexpression in the perimenopausal myometrium. The statistical test applied was a MAST test with p-values corrected for multiple comparisons by FDR. **E** Representative refined spatial maps of contractile capillary cells in the perimenopausal (left) and postmenopausal (right) myometrium ($n = 3$ peri and 5 post-menopausal samples). Colors represent the scaled proportion of this cell type in each location (red indicates the highest proportion of contractile capillary cells in that tissue). **F** Boxplot of the scaled cell proportions of contractile capillary per spot split in the postmenopausal and perimenopausal myometrium ($n = 3$ peri and 5 post-menopausal samples; the center line shows the median for the data and error bars extend to the largest and smallest value no further than 1.5 inter-quartile range; Unpaired two-sided Wilcoxon test where ***$p$ value < 2.2e−16). Nhood size Neighborhood size. Source data are provided as a Source Data file.

SMC subtype expressed classical markers, such as *ACTG2, MYL9*, and *TPM2* (Supplementary Data 7). Contractile SMCs expressed contractility-associated genes (e.g., *CARMN, KANSL1, PBX1* and ion channel genes (e.g., *KCNMA1, CACNA1C*, and *CACNA1D*). Stimuli-response SMCs expressed *LMCD1, HRH1, MAFF, SAMD4A*, and *HOMER1*, which participate in inflammatory responses. Finally, stressed SMCs expressed stress-associated genes (e.g., *JUN, JUNB, FOSB, ZFP36*, and *GADD45B*).

Comparative analyses revealed a significant increase in the abundance of contractile and canonical SMCs (Fig. 4B, C) and the reduced expression of contractility-associated genes (e.g., *SMTN, ACTG2, MYL6, MYL9, TPM2*, or *DES*) in all SMCs during myometrial aging (Supplementary Fig. 4A, Supplementary Data 8). Interestingly, we found a significant downregulation in K+ voltage channel gene expression (*VDAC1, VDAC2, KCTD12, KCNIP4, KCNE4*) in the postmenopausal myometrium (Fig. 4D). Spatial transcriptomics analysis showed an increase in the proportion of SMCs in perimenopause, which was only detectable in the distal section of the tissue, furthest from the myometrium (Supplementary Fig. 4B, C). Conversely, it confirmed a reduction in the expression of contractility marker genes such as *SMTN* (Supplementary Fig. 4D, E) and voltage channel genes such as *KCNE4* (Fig. 4E), also confirmed at protein level (Fig. 4F).

Our results suggest that a greater SMC abundance with lower contractile and ion-conductive activities accompanies myometrial aging.

### Age-related inflammation/ DNA damage in perivascular cells

Analysis of PV cells in the myometrium revealed four pericyte and three vascular smooth muscle cell (VSMC) populations (Fig. 5A, Supplementary Data 9). We identified pericyte subtypes based on the expression of canonical markers *RGS5* and *PDGFRB*[18] The RGS5⁻ PDGFR⁻ subtype (contractile pericytes) presented a partial loss in canonical markers but a gain in the expression of contractility markers such as *DES* and *ACTG2*. VSMC subtypes included contractile VSMCs expressing *ACTA2, ELN, CARMN*, and *KCNMA1*, damage-responsive VSMCs expressing stress and DNA repair-associated genes (*ATF3, FOS, JUN, IER2, KLF2*), and stimuli-responsive VSMCs expressing genes related to the transmission of impulses (*ATP1B3* and *TNC*), arterial damage (*PVT1*), systemic inflammation (*CCL2*) and transcription control (*CCNH* and *ZNF331*).

Analysis of the top ten differentially expressed genes in PV cells revealed an association between myometrial aging and the upregulated expression of genes related to inflammation, immune response, and DNA damage, but the downregulated expression of contractility genes associated with Ca2+/K+-related signaling and vascular health-related genes (*MGP, TAGLN, ACTA2, ACTB, ACTG1, APOE*, or *KCND2*) (Fig. 5B, Supplementary Data 10). Spatial transcriptomic analysis revealed a decreased co-localization of endothelial and PV cells during myometrial aging (Fig. 5C), suggesting an age-related impediment in the endothelial compartment support by PV cells.

Our findings indicate that myometrial aging prompts a decline in the PV cells that support the endothelial compartment, which may adversely affect optimal endothelial function.

### Monocytic dysfunction as a hallmark of the aging myometrium

Finally, we analyzed ten different subpopulations of immune cells in the aging myometrium (Fig. 5D). Myeloid cells encompassed four subclusters (neutrophils, monocytes, dendritic cells, and macrophages), while lymphoid cells encompassed five subclusters (B-cells, T-cells, CD4 + T-cells, CD8 + T-cells, and NK-cells) (Fig. 5E; Supplementary Data 11).

Although we did not find changes in immune cell abundance, differential gene expression analysis revealed significant differences, mainly in monocytes (Supplementary Data 12), which overexpressed genes implicated in inflammatory processes and in the facilitation of immune responses through monocyte activation, adhesion, and migration (e.g., *PBX1, BNC2, EGR1, RORA*, and *DCN*). These genes were enriched in pathways associated with interleukin signaling, T-cell response modulation, and macrophage activation (Fig. 5F). Our results suggest that myometrial aging affects the expression of genes responsible for maintaining the innate immune response through T-cell activity and macrophage activation through cytokine signaling in monocytes.

### Altered cell-to-cell communication in aging myometrium

We next examined the impact of aging on CCC within the aging myometrium. The postmenopausal myometrium possessed less intricate signaling networks than the perimenopausal myometrium and shifts in ligand-receptor pair interactions (Fig. 6), which suggests that reduced CCC accompanies myometrial aging.

We identified 204 pathways in perimenopausal and postmenopausal myometria and highlighted potentially relevant pathways to myometrial function during aging. These include the fibrosis-associated platelet-derived growth factor (PDGF) and insulin growth factor (IGF) pathways, the angiogenesis-associated angiopoietin-like (ANGPTL) and endothelin (EDN) pathways, the inflammation-associated C-X-C Motif Chemokine Ligand (CXCL) pathway, and the neurexin (NRXN) pathway, with a role in impulse transmission (Supplementary Fig. 5A, B).

In the aging myometrium, PDGF signaling suffered a notable reduction in outgoing signals from PVs and SMCs (Fig. 6A), while we also observed a shift in the relative contribution of specific ligand-receptor pairs such as PDGFC-PDGFR (associated with fibrosis and inflammation) (Fig. 6B). We discovered an age-related increase in signals emanating from fibroblasts and SMCs for the IGF pathway (Fig. 6C) characterized by the loss of IGF2-associated interactions and a subtle reduction of IGF1-IGF1R interactions (Fig. 6D).

ANGPTL pathway alterations affected PV cells, endothelium, and SMC subtypes, with their decline suggesting reduced blood vessel sprouting during aging (Fig. 6E), while a shift from EDN signaling in the

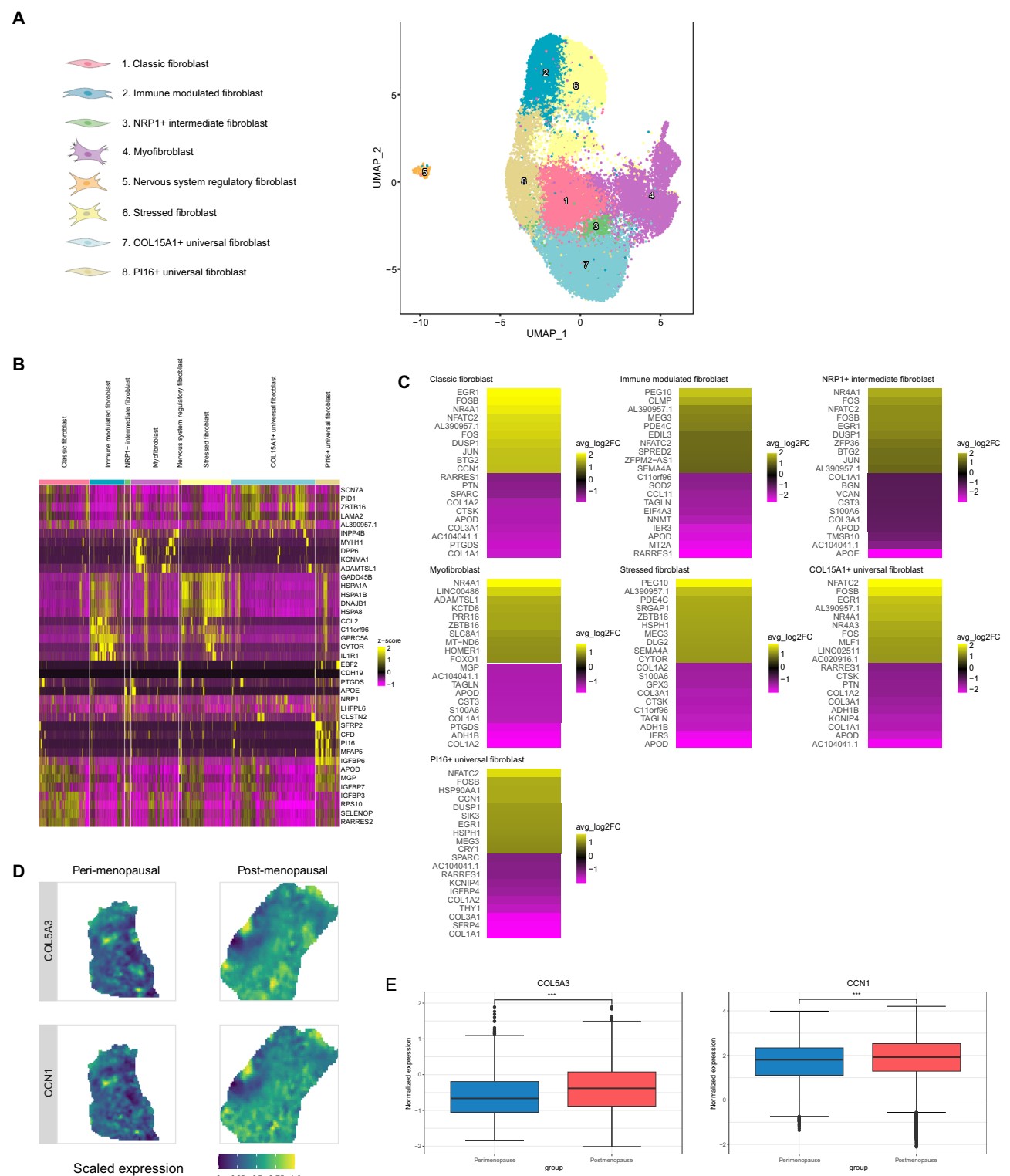

**Fig. 3 | Age-related changes in the myometrial fibroblast population. A** UMAP visualization displaying the fibroblast subpopulations in the myometrium at 0.6 resolution. **B** Heatmap showing the relative expression of the top five discriminatory genes in each subpopulation. **C** Heatmap showing the genes with the top ten highest and lowest LogFC values in the postmenopausal vs. perimenopausal myometrium for each subpopulation. Positive LogFC indicates overexpression in the postmenopausal myometrium, whereas negative LogFC indicates overexpression in the perimenopausal myometrium. **D** Representative refined spatial maps of the expression of *COL5A3* and *CCN1* in the perimenopausal (left) and postmenopausal (right) myometrium (*n* = 3 peri and 5 post-menopausal samples). Color indicates expression levels in each spot (yellow indicates the highest expression of each gene in the tissue). **E** Boxplot of the spatial expression of the collagen *COL5A3* (left) and fibroblast senesce marker *CCN1* (right) genes in the perimenopausal and postmenopausal myometrium (*n* = 3 peri and 5 post-menopausal samples; the center line shows the median for the data and error bars extend to the largest and smallest value no further than 1.5 inter-quartile range; Unpaired two-sided Wilcoxon test where ***$p$ value < 2.2e−16). Source data are provided as a Source Data file.

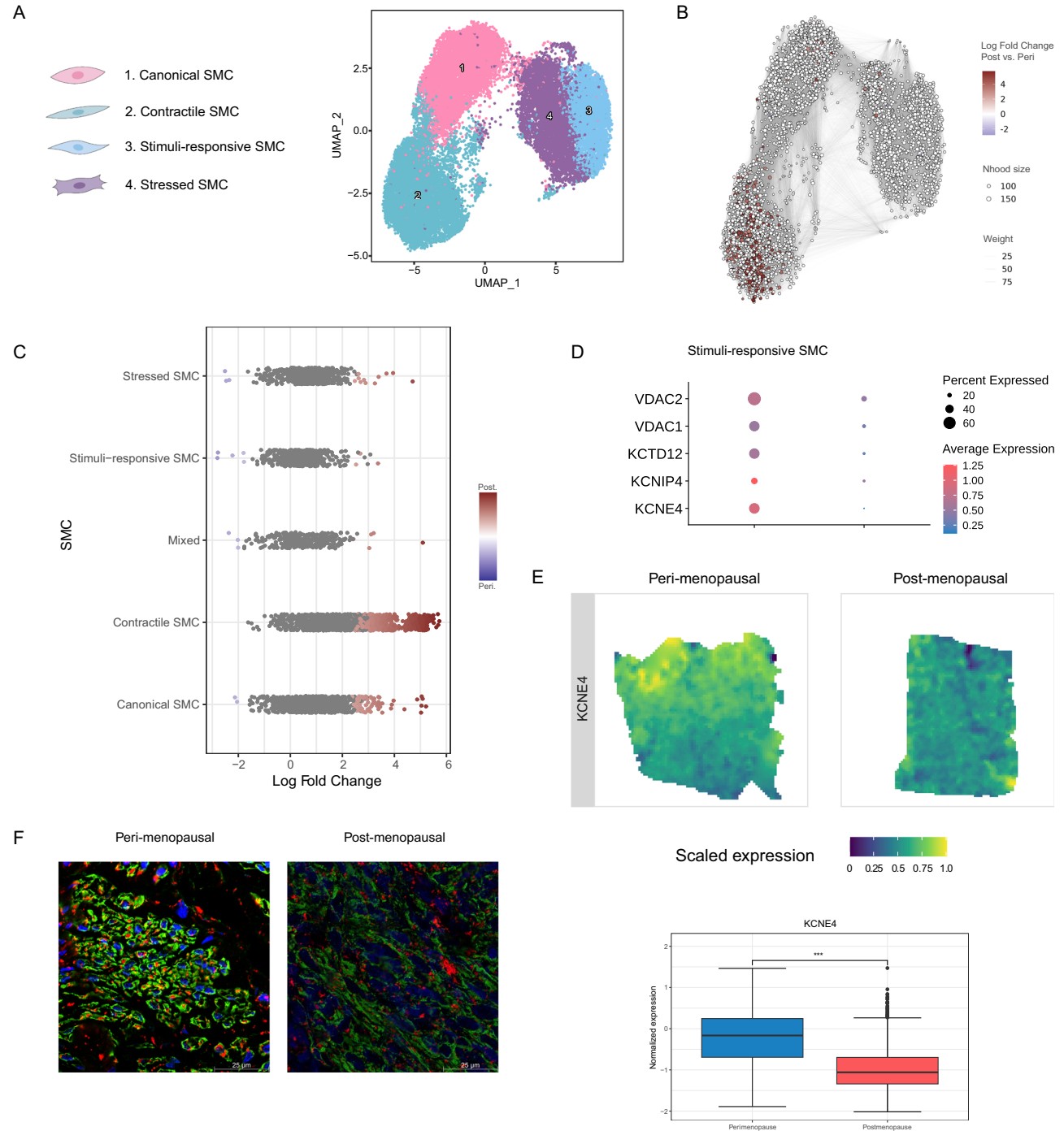

**Fig. 4 | Increased abundance but reduced functionality of SMCs accompanies myometrial aging. A** UMAP visualization of the four distinct SMC subpopulations. **B** Neighborhood graph highlighting the differential abundance of SMCs in the aging myometrium. Neighborhoods colored in dark red represent those with significantly increased abundance in the postmenopausal myometrium at 0.6 resolution. **C** Beeswarm plot of differential SMC abundance by cell type. **D** Dot plots representing the differential gene expression of voltage channel encoding genes during myometrial aging in stimuli-response SMCs. Dot size indicates the percentage of stimuli-response SMC that express the gene, while color indicates average expression. **E** Spatial expression (top) and boxplot (bottom) of potassium voltage-gated channel gene *KCEN4* in representative refined spatial maps of perimenopausal (*n* = 3; left) and postmenopausal (*n* = 5; right) myometrium, where color indicates expression levels in each spot (the center line shows the median for the data and error bars extend to the largest and smallest value no further than 1.5 interquartile range; Unpaired two-sided Wilcoxon test where ***\*\*\*p* value < 2.2e−16). **F** Representative inmunofluorescence image of ACTA2 (green) and VDAC1/2 (red) in peri (*n* = 3; left) and postmenopause (*n* = 3; right). Scale bar = 25 μm. SMC smooth muscle cells, Nhood size Neighborhood size, ACTA2 Actin Alpha 2, VDAC1/2 Voltage-dependent anion-selective channel 1/2. Source data are provided as a Source Data file.

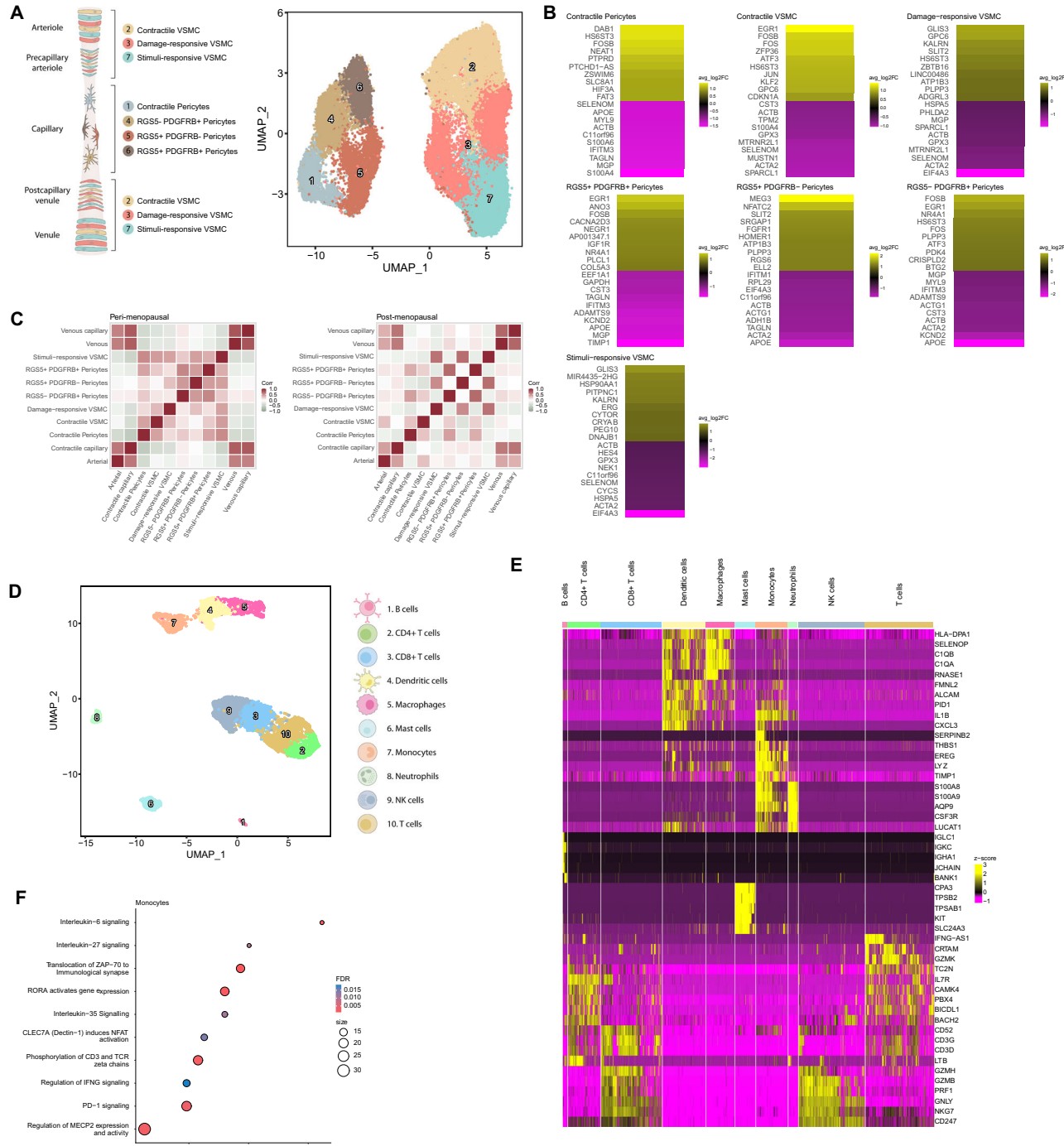

**Fig. 5 | Impairments in the Myometrial Perivascular and Immune Cell Populations Associated with Myometrial Aging. A** Schematic representation of PV subtypes surrounding the myometrial microvasculature from the arteriole to the venule: arterioles and venules are surrounded by a single layer of contractile VSMCs, while pericytes (characterized by a stellate shape) are usually found surrounding smaller and transitional vessels such as precapillary arterioles and postcapillary venules. Contractile pericytes provide support in capillaries (left). UMAP visualization displaying the PV subpopulations at 0.8 resolution (right). **B** Heatmap showing the genes with the top ten highest and lowest LogFC values in the postmenopausal vs. perimenopausal myometrium for each PV subpopulation. Positive LogFC indicates overexpression in the postmenopausal myometrium, whereas negative LogFC indicates overexpression in the perimenopausal myometrium.

**C** Correlation of the spatial distribution of endothelial and PV cell populations in the perimenopausal (left) and postmenopausal (right) myometrium, with color scaled by correlation value (darker red indicates a higher correlation between the location of the indicated cell types). **D** UMAP visualization displaying the immune cell sub-clusters present in the myometrium at 0.5 resolution. **E** Heatmap showing the relative expression of the top five discriminatory genes in each subpopulation. **F** Dotplot showing the representative biological processes and pathways affected in monocytes during myometrial aging based on differential gene expression. Significant over-representation of biological processes and pathways (color intensity) shown by each gene set (dot size) from perimenopausal and postmenopausal monocytes. VSMC vascular smooth muscle cells, NK natural killer, FDR false discovery rate.

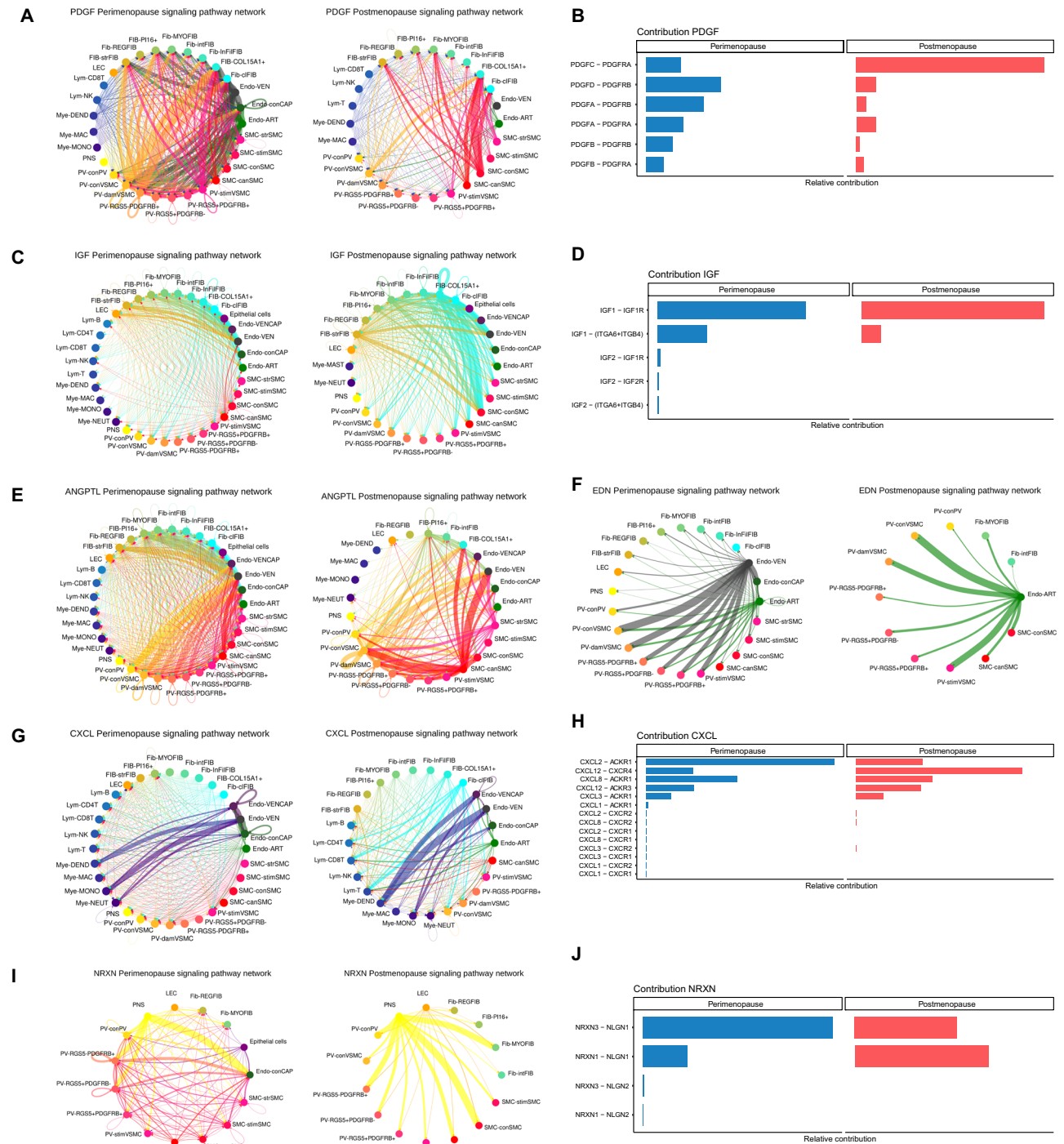

**Fig. 6 | Age-related changes in myometrial cell-to-cell communication.** Arrows between cell types are colored according to the cell type emitting the signal. The relative thickness of each line depicts the expression-based strength of the interaction between cell types. CCC chord plots for **A** PDGF, **C** IGF, **E** ANGPTL, **F** EDN, **G** CXCL and **I** NRXN signaling pathways in the perimenopausal (left) and postmenopausal myometrium (right). The relative contribution of ligand-receptor pairs to CCC in the **B** PDGF, **D** IGF, **H** CXCL, and **J** NRXN signaling pathways in the perimenopausal (left) and postmenopausal myometrium (right). PDGF platelet-derived growth factor, IGF insulin growth factor, ANGPTL angiopoietin-like, EDN endothelin, CXCL C-X-C Motif Chemokine Ligand, NRXN nerve transmission-associated neurexin, Fib fibroblasts, Endo endothelial, SMC smooth muscle cells, VSMC vascular smooth muscle cells, PV perivascular, LEC lymphatic endothelium, Mye myeloid, Lym lymphoid, VEN venous, ART arterial, str stressed, stim stimuli-response, con contractile, can canonical, dam damage-response, MAC macrophages, DEND dendritic, NK natural Killer, MONO monocytes, NEUT neutrophils, REG nervous system regulatory fibroblast, MYO myofibroblast, int NRP1 intermediate, Inf immune modulated fibroblast, Cl classic.

arterial and venous endothelium to only the arterial endothelium also accompanied myometrial aging (Fig. 6F).

We encountered notable alterations to immune system associated CXCL signaling; we observed a shift from extensive communication between various cell types to stronger communication of fibroblasts with immune cells and myeloid cells with endothelial subtypes during myometrial aging (Fig. 6G). We observed a shift from the activity of the CXCL2-ACKR1 ligand-receptor pair in perimenopausal myometria to CXCL12-CXCR4 in postmenopausal myometria (Fig. 6H).

Lastly, analysis of the impulse nerve transmission associated NRXN signaling pathway suggested a shift to fewer but stronger interactions, particularly from the PNS to PV, SMC, and fibroblast subpopulations during myometrial aging (Fig. 6I). Moreover, we identified a change between the NRXN3-NLGN1 and NRXN1-NLGN1 ligand-pairs, which decreased and increased during aging, respectively (Fig. 6J).

Interestingly, CCC belonging to several signaling networks (ANGPTL, CXCL, and PDGF) among major cell types (SMC, fibroblasts, VSMC, and endothelium) were validated at tissue level, locating the direction and strength of each interaction (Supplementary Figure 6).

Overall, CCC data reveals a generalized decline in signaling complexity accompanying myometrial aging, which may prompt detrimental processes such as fibrosis, inflammation, impaired angiogenesis, and reduced responsiveness to stimuli. Given the limitation when deciphering cell responses to steroid hormones through the CCC using CellChat[16], we also evaluated the expression of estrogen receptors 1 and 2 (ESR1/2) and progesterone receptor (PGR) in our data. While no differential expression affected ESR2, ESR1 appeared upregulated in all cell subpopulations in postmenopause, as our results at single cell transcriptomics and spatial level show (Supplementary Fig. 7A, C, E). Conversely, PGR appeared downregulated only in contractile SMC in postmenopause as according to the DE results, which was also addressed by spatial representation (Supplementary Fig. 7B, D, F). Results for ESR1 and PGRβ were validated at protein level through immunofluorescence assays (Supplementary Fig. 7G–J).

### The loss of signaling pathways accompanies myometrial aging

CCC analysis revealed that myometrial aging associated with the loss of interactions from 25 signaling pathways (of 229), which encompassed functions such as angiogenesis (HGF), homeostasis and tissue repair (CALCR, CLDN, PERIOSTIN, and BMP), contractility (ncWNT), immune processes (CD70, IL10, FASLG, SEMA7, CD48, CD22, CSF3, and CHEMERIN), and nervous system regulation (L1CAM and NGF) (Supplementary Data 13).

The loss of interactions driven by regulatory fibroblasts to endothelial cells through HGF signaling suggests impaired angiogenesis during aging (Fig. 7A). Regarding tissue repair/homeostasis (Fig. 7B), we observed the loss of CALCR and PERIOSTIN signaling postmenopause - CALCR signaling functioned between myeloid cells and stressed fibroblasts in the perimenopausal myometrium, while PERIOSTIN signaling occurred through signals originating from stimuli and contractile SMCs. We also found the loss of CLDN signaling, which was driven mainly by PI16+ fibroblasts in the perimenopausal myometrium. We also discovered the loss of non-canonical Wnt (nc-WNT) signaling, required for SMC contraction, during aging[19] (Fig. 7C). This pathway primarily involved signals outgoing from contractile SMCs and arterial endothelium in the perimenopausal myometrium. For immune system-related pathways (Fig. 7D), we discovered the loss of anti-inflammatory IL10 signaling between myeloid cells and B-lymphocytes. Similarly, the CD22 pathway, which negatively regulates B-cells and involves signaling from mast cells, monocytes, and B-cells to other immune cells, was lost during myometrial aging. Lastly, we observed the loss of CD48 signaling, which exhibited strong autocrine signaling in NK lymphocytes in the perimenopausal myometrium.

We discovered the loss of pathways associated with nerve stimulation and transmission during myometrial aging, specifically through the NGF and L1CAM pathways (Fig. 7E). The NGF pathway originated from PV cells, fibroblasts, stressed fibroblasts, and SMCs responsive to stimuli, but extended to PNS and nervous system-regulatory fibroblasts (Fig. 7E). Meanwhile, the L1CAM pathway predominantly transmitted signals from PNS to immune cells.

These findings highlight the complete loss of specific CCC networks in the aging myometrium, impacting pathways involved in angiogenesis, tissue repair, contractility, immune processes, and nervous system regulation.

Altogether, our work provides a detailed atlas of the human myometrium and reports an age-related reduction in the proportion of contractile cells and the impaired expression of genes in all cell types, such as voltage channels and contractility genes in SMCs and endothelial cells, ECM-related genes in fibroblasts, and inflammation-associated genes in immune cells. Furthermore, myometrial aging disrupts interactions between cell types, resulting in defective angiogenesis, tissue repair, immunity, and nervous system regulation.

## Discussion

Aging represents a complex process encompassing altered tissue repair, fibrosis, tissue reprogramming, and cellular senescence[20]; with a systemic impact including reproductive organs. While previous single-cell transcriptomic atlas of human uterus[12] have been earlier described, also including those related to human parturition[13], and fibroid-free myometria[14], our study uses a perimenopause versus postmenopause comparison as a model system, which offers a description of the cellular network present in the human myometrium and provides evidence for altered CCC as a hallmark of myometrial aging, addressing a notable gap in previously published data. We document age-related endothelial dysfunction, which may contribute to microvasculature remodeling and impaired angiogenesis, and report an age-related reduction in responsiveness to stimuli/electrical cues and disrupted signaling involved in SMC contraction. Finally, age-induced fibrosis and perturbation to the immune system may prompt compromised reproductive and obstetric function in older individuals.

We generated a comprehensive cellular landscape of the aging myometrium, which revealed the transcriptomic activity of 23 myometrial cell subpopulations, including contractile and venous capillary cells, immune-modulated[21], stressed, and nervous system regulatory fibroblasts[22], stimuli-responsive SMCs, and damage-/stimuli-responsive VSMCs. Overall, these findings highlight the cellular heterogeneity of the aging myometrium. When studying endothelial cells, we discovered an age-related reduction in the proportion of contractile capillary cells in postmenopause; however, we also observed altered function, as revealed by the downregulated expression of contractility genes and the upregulated expression of genes linked to senescence and extracellular matrix deposition[23]. Moreover, we detected the upregulated expression of aging-associated genes in universal PI16+ and COL15A1+, classic, and intermediate NRP1+ fibroblasts during myometrial aging. SMCs also exhibited reduced functionality, as shown by the downregulated expression of contractility-associated genes despite an increase in their proportion, which may represent an attempt to compensate for the age-related loss of functionality[24]. Stimuli-response SMCs in the aging myometrium downregulated their expression of *KCNE4*, a type I beta subunit that modulates the channel's gating kinetics and stability[25] and *VDAC2*, which has been linked to cardiac muscle atrophy in mice[26]. Spatial transcriptomics analysis revealed a change in the SMC proportion between peri and postmenopause that was only appreciable in the distal region of the myometrium from the endometrium. These findings are in line with previous reports highlighting the relevance of the endometrial-myometrial junctional zone in reproductive outcomes and changes in this region that take place with the aging process[27]. The immune system plays a crucial role in senescence and aging, leading to chronic inflammation (inflammaging) that disrupts cell/organ function[28]. Indeed, the process of myometrial senescence, carried by decreased proliferation and dysregulated gene expression in senescent cells[29,30] is patent in our results through an upregulation of genes such as *JUN* and *FOS*. We observed the upregulated expression of genes related to immune cell activation, adhesion, migration, myelopoiesis, and monopoiesis in the aging myometrium, suggesting disrupted immune homeostasis.

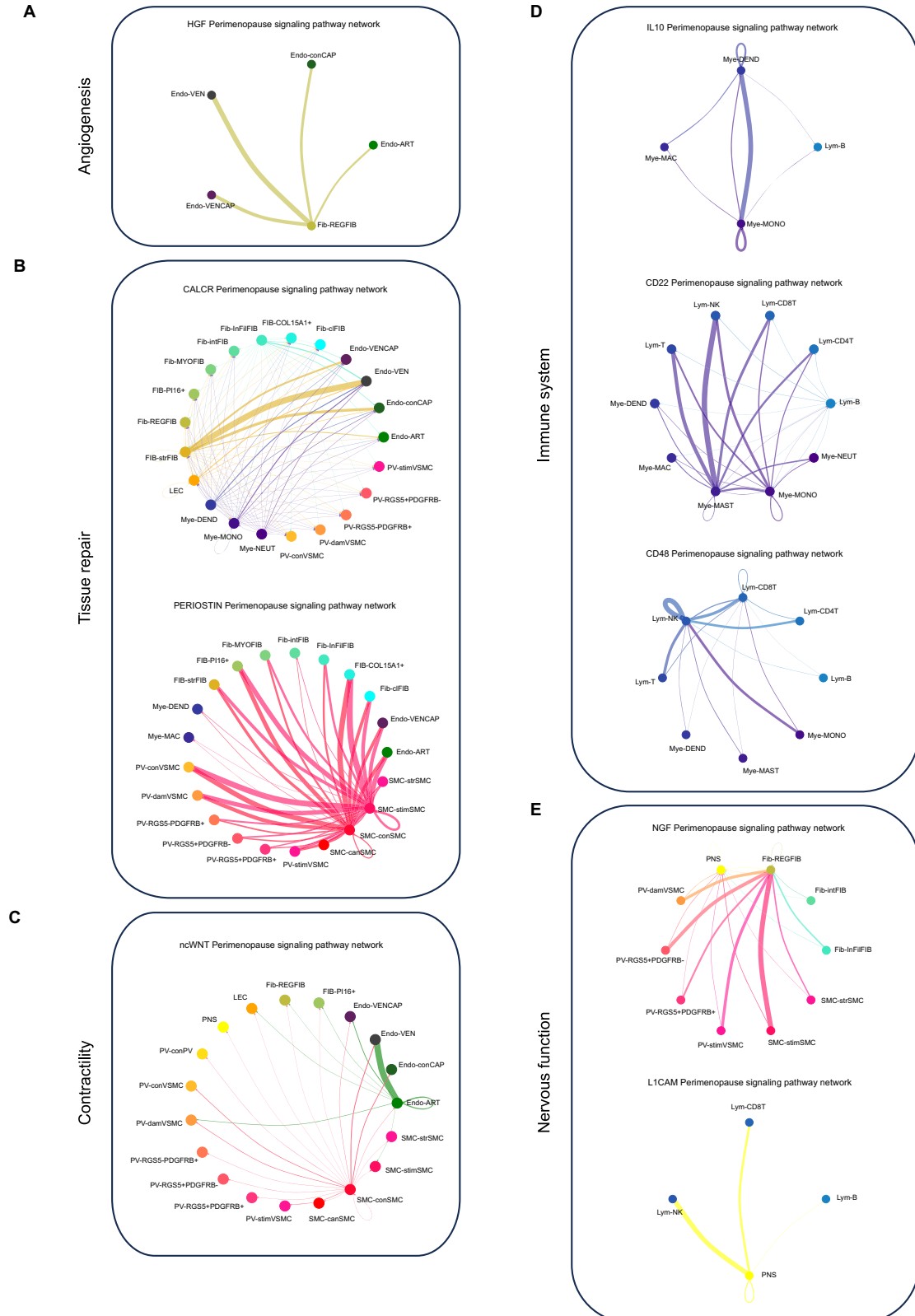

**Fig. 7 | Lost Signaling Pathways in the Aging Myometrium.** CCC chord diagrams displaying signaling pathways active in the perimenopausal myometrium but lost during aging, including those related to **A** angiogenesis (HGF), **B** homeostasis and tissue repair (CALCR, PERIOSTIN), **C** contractility (ncWNT), **D** immune processes (IL10, CD22, CD48) and **E** nervous system regulation (NGF, L1CAM). Arrows between cell types show the direction of the interactions and follow the color code of cell types commonly detected in menopause. The relative thickness of each line depicts the expression-based strength of the interaction between cell types.

Our findings confirmed age-related alterations to CCC in the myometrium, which may prompt disturbances in interactions and physiological processes frequently observed as age-related manifestations[31]. In fact, the loss of CCC pathways outlined in our study highlights these pathways as key contributors to the age-related decline in the functioning of angiogenesis, immune responses, and tissue repair. This decline is attributed to a reduction in the "beneficial" intercellular communication[32] and may have broader applicability to other organ systems. Interestingly, we discovered the age-related loss of HGF and NGF signaling, which function in angiogenesis and blood flow. This loss could reduce HGF-mediated wound-healing and anti-apoptotic/anti-inflammatory effects; meanwhile, lost NGF signaling could impact physiological angiogenesis through interactions with VEGF[33,34].

Pathways promoting angiogenesis (e.g., ANGPTL, ANGPT, and VEGF[35]) also displayed reduced activity during myometrial aging; we additionally discovered changes in the EDN pathway, which regulates blood flow through vessel constriction or relaxation. The predominant Endothelin Receptor Type A (EDNRA) ligand found postmenopause has been linked to aging[36]. Alterations to blood flow regulation and impaired neovascularization, in compliance with previously reported endothelial issues such as myometrial artery calcifications[37] could impact the myometrium's ability to control hemorrhage and nutrient delivery, contributing to increased morbidity/mortality in older individuals during pregnancy and labor[38].

As previously reported in other models like mice, several pathways are dysregulated, resulted in collagen-deposition disruptions[39]. Our study revealed a significant enrichment of fibrosis-related pathways during aging, which may compromise myometrial function. We observed increased PDGFC (implicated in fibrosis) and reduced PDGFD interactions (associated with angiogenesis and blood vessel maturation)[40,41]. The IGF pathway exhibited subtle alterations, with stronger interactions between IGF1 and IGF1R (involved in pulmonary fibrosis[42]) in postmenopause.

We detected changes in interacting ligand-receptor pairs of the NRXN signaling pathway, interfering with impulse transmission and SMC contraction[43]. The simplification/depletion of such CCC networks in the aging myometrium may cause a decline in nerve impulse transmission and decreased contractility essential for pregnancy and labor. Our findings suggest that the decreased contractile potential during myometrial aging may justify why older parturients have longer labors[44] and require higher dosages of oxytocin for third-stage labor to prevent uterine atony[45].

Our analyses revealed the absence of the ncWNT pathway in postmenopause, which participates in tissue homeostasis and cell migration[46], as *WNT5A* participates in cardiac muscle contraction[19] and strengthens actin cytoskeleton assembly[47]. We observed a loss of CALCR signaling, required for maintaining muscle stem cells in a quiescent state[48]. Myometria unresponsive to stimuli/contractions can lead to uterine atony, resulting in postpartum hemorrhage[38] or labor dystocia.

Finally, we discovered the absence of pathways involved in immune homeostasis (such as IL10[49], CD22[50] and CD48[51]) in the postmenopausal myometrium. We also observed significant changes in the CXCL pathway; while ACKR1 protects against inflammaging by limiting immune cell extravasation[52], CXCR4 enhances the inflammatory response associated with age-related DNA damage[53]. These changes suggest that impaired immune autoregulation in the aging myometrium may prompt an overactive immune system. Inflammatory status determines birth at term/preterm, as cytokines released by immune cells regulate contractility leading to parturition[54] while CXCL12 produced by the myometrium represents a causative factor for preterm labor[55]. Attenuating exacerbated myometrial immune responses may represent a therapeutic avenue to reduce the incidence of preterm birth in older individuals[56].

Notably, the expression patterns of ESR1 might indicate a compensatory mechanism in response to the declining levels of estrogen as individuals age, enhancing their responsiveness to estrogens. ESR1 has been previously implicated in various aging-related processes, including vascular health[57] and it plays a crucial role in maintaining myometrial quiescence during pregnancy[58] underscoring its significance in myometrial physiology. Although PGR is widely recognized as a regulator of myometrial contractility, isoforms α and β have opposite functions, as isoform α promotes uterine contraction while isoform β induces relaxation[59]. Our immunofluorescence results validated the downregulation of PGRβ, providing further evidence that changes in contractility are a significant characteristic of aging in the myometrium, corroborating results obtained by spatial transcriptomics.

Assembling all results, several populations exhibited increased susceptibility to the aging process during post menopause, as they display the highest number of DE genes and significant alterations in CCC patterns. Specifically, stimuli-responsive SMC, contractile SMC, canonical SMC, perivascular cells including RGS5+PDGRFB+ and RGS5+PDGRFB- pericytes and stimuli-responsive VSMC and stressed and myofibroblasts exhibited over 1000 DE genes. CCC differences between peri and postmenopause revealed stimuli-responsive SMC, canonical SMC, immune-modulated fibroblasts, and stimuli-responsive VSCM experienced the most significant shifts, aligning with the DE results, and extrapolating these hallmarks of aging from a molecular to a biological ground.

We acknowledge several limitations associated with our study. First, bias in cell/nuclei recovery could impact the likelihood of capture rates, for instance due to i) the size of certain cell types that may cause them to break when entering the microfluidic system. This could explain why SMCs were not the majoritarian cell type detected in our work despite the myometrium being a mainly muscular tissue, due to the increased size of SMCs[13]; ii) Furthermore, it is possible that our study may not have sufficient cells to accurately detect and characterize stem/progenitor cells due to the low proportions within adult human tissues[60]. Second, the resolution of spatial transcriptomics did not reach the level of individual cells, which could hinder the visualization and estimation of minor cell populations, which could have also influenced the integration of spatial transcriptomics with CCC data. Further, these two limitations may explain why we report differences in the proportion of SMCs, which was higher according to scRNAseq, and lower according to spatial transcriptomics. We attribute this apparent inconsistency to the lack of spatial context of scRNAseq, as we found differences in SMC content depending on the region (distal or proximal) that cannot be considered in the analysis of single cell. We also acknowledge that spatial transcriptomics samples were limited to the junctional zone of the myometrium, which may have radically different molecular and cellular features[61] than the rest of the myometrium (middle and outer layers) which was studied by scRNAseq. Nevertheless, integrating sc/snRNA-seq and spatial transcriptomics data represented the most appropriate approach for uncovering cell diversity and spatial gene expression patterns. Third, the choice of stablishing two age groups (peri- vs. postmenopause) instead of considering the age as a continuum[8,11] has been made due to the distribution of ages in the patients recruited for the study, which is limited due to the difficulty of the myometrium to get biopsied, especially in younger patients and those that have not undergone hysterectomies. Notwithstanding, putative changes that arise as a continuum of the aging process would be anyway tangible when comparing two different age groups. In an effort to address data gaps from younger individuals, we've consulted other myometrium atlases. Nevertheless, these references diverge from our study in examining myometria without fibroids[14], during parturition[13], and by treating the uterus as a whole without specific emphasis on the myometrium[12]. Lastly, it is important to acknowledge that this study is focused only on the

myometrium, but an integral approach considering other uterine tissues, other organs and other age-associated conditions (hypertension, diabetes, cancer…) is necessary to understand the complex mechanisms implied in poorer pregnancy outcomes that happen with increased maternal age. In this regard, this study, and contrary to what occurs in others[8,11] that eliminate confounders, the patients recruited may have had previous deliveries or other health conditions such as age-associated conditions that may affect the local changes we see at the myometrial tissue level.

Among the strengths of this work, we first highlight the extensive dataset of cells we present, totaling 161,202 cells and 24,918 nuclei. This number is significantly higher if compared to other atlases: 53,194 cells of myometria in human parturition[13], 96,573 cells in leiomyoma-free myometria[14], and 7124 cells (only considering the uterus) in the Tabula sapiens[12]. On the other hand, the work includes two cutting-edge study techniques: sc/sn-RNAseq and spatial transcriptomics. Even though each of them independently provides valuable insights in the human myometrial architecture and gene expression, is the integration of both which leverages the study.

Overall, our findings provide a descriptive study of the human aging myometrium at single-cell level, revealing diminished contractility, impaired angiogenesis, increased fibrosis and inflammation as the main drivers of senescence in the myometrial tissue. Such traits could explain impaired peristalsis events and aberrant contractility in aged women, which support a role of the myometrium in dystocia[62], uterine atony[63], preterm labor[9,54], or postpartum hemorrhage[64] and set the basis for further works providing experimental validation and additional comparable groups.

Early detection and management of these potential obstetric complications can help mitigate the risks associated with the increasing prevalence of advanced maternal age pregnancies, as well as to the design of preventive strategies to reduce these labor complications in aged women.

## Methods

All experimental procedures and bioinformatic analyses in this work comply with ethical regulations and good scientific practices.

### Participant details

This prospective, multicenter, descriptive case series included twenty females, both living and brain death donors, with ages ranging from 46 to 79 years old. Our research employs the term "women" to refer to individuals with a uterus that undergo menopause while acknowledging that not all individuals identifying as women have a uterus and/or experience menopause, and not all individuals undergoing menopause identify as women. Patients were divided into two groups: the perimenopausal group (46–54) and the postmenopausal (>54). In accordance with the definition of menopause as the absence of menstruation for a minimum of 12 consecutive months, our assumption, supported by prior studies[4] and patients' clinical reports, was that individuals aged 55 and above were postmenopausal. Conversely, patients aged 46–54 were categorized as perimenopausal due to the likelihood of hormonal fluctuations and unpredictable menstrual cycle patterns during this phase. Supplementary Data 1 describes the demographic and clinical characteristics of patients.

All procedures involving human tissue samples were approved by the Institutional Review Board of the Spanish hospitals involved: Hospital Clinico Universitario, Valencia, Spain (November 5th, 2019); Hospital La Fe, Valencia, Spain (December 4th, 2019); Hospital General Universitario, Valencia, Spain (February 12th, 2021). The Hospital La Fe provided fifteen uteri from women undergoing hysterectomy for pelvic prolapse, while a total of five uteri (two from Hospital Clinico Universitario and three from Hospital General) were obtained from patients with brain death under the Organ Donor Program with non-

cancer-related causes or traumatic injury. These five uteri were acquired through the surgical extraction of the entire uterus after meeting the criteria for brain death, following the protocol approved by the ethics committee. This approach permits the recovery of organs before circulatory arrest, thus minimizing the duration of warm ischemia. The surgical removal of the uterus was executed using a method of static cold preservation, in the same planned surgery for the removal of the organs to be transplanted. Subsequently, the uterus was submerged in a preservation solution and transported within a temperature range of 3–5 °C. This procedure mirrors the process for uterus transplant, which indeed is the same we followed for uteri obtained from hysterectomies conducted due to prolapse. All patients and donor families provided written informed consent, and those with gynecological disorders such as endometriosis, uterine malformations, uterine leiomyoma, endometrial polyp, hyperplasia, uterine septum, Asherman's syndrome or hydrosalpinx, were excluded from the study. Patients were not considered for inclusion in the study if they had previously been diagnosed with malignancies related or unrelated to the uterus, and if such malignancies were also identified during the surgical procedure. To mitigate the potential effects of previous infections, inclusion in the study also necessitated negative serological tests for HIV, HBV, HCV, and RPR. This study was conducted per the International Conference on Harmonization Good Clinical Practice guidelines and the Declaration of Helsinki. All specimens were anonymized after collection and histologically evaluated by board-certified pathologists to confirm the diagnosis according to World Health Organization criteria. Neither the participants nor any of their relatives obtained any kind of economical compensation for their participation on this study.

### Tissue collection and sample preparation

Uterine tissues from the twenty recruited patients ($n = 3$ samples per patient) were maintained in preservation solution (HypoThermosol® FRS, Stemcell technologies, #07936) after surgery and further dissociated within 24 h of tissue retrieval to isolate the myometrial layer, which was achieved by removing the endometrium, serosa, and necrotic areas with a sterile scalpel.

As previously described[65], myometrial samples were rinsed in a wash buffer solution containing Hank's Balanced Salt Solution (HBSS) (ThermoFisher Scientific, Gibco™ #14025) and 1% antibiotic-antimycotic solution (ThermoFisher Scientific, Gibco™ #15240112) to remove blood and mucus. Samples were then divided for different purposes; snap-frozen in liquid nitrogen or formalin fixation/paraffin embedding (FFPE) for further histological characterization, while the remaining tissue was carefully manually minced into small pieces (<1 mm³) and digested at 37 °C using an enzymatic process for single-cell dissociation. Subsequently, cell suspensions were passed through a 50-μm polyethylene filter (Partec, Celltrics, #04-004-2327) to remove cell clumps and undigested tissue and then dissociated to single cells by incubating with 400 μL TrypLE Select (ThermoFisher Scientific, #12563029) for 20 min at 37 °C to obtain single-cell suspensions. 100 μL DNase I (Sigma-Aldrich, #D4513) was then added to digest extracellular genomic DNA, and cells were treated with ACK Lysing Buffer (ThermoFisher Scientific, #A1049201) to induce hypotonic shock to avoid red blood cell contamination. The resulting cell suspension was pipetted, passed through a 100-μm cell filter, and centrifuged at $500 \times g$ for 5 min. The cell pellet was resuspended in Dulbecco's Modified Eagle's Medium (DMEM, Sigma-Aldrich #D5921) supplemented with 2% fetal bovine serum (Capricorn Scientific, #FBS-11A) and 10 mM HEPES (ThermoFisher Scientific, #15630080) as a myometrial-enriched suspension at a concentration of $1 \times 10^6$ cells/mL. Cell concentration and viability were measured using trypan blue with an EVE™ automated cell counter (NanoEnTek). Dead cells were removed with the Dead Cell Removal Kit (Miltenyi Biotec, #130-090-101) cells, with cell suspensions reaching a viability of ≥70%.

## Nuclei isolation for snRNA-seq from myometrial cell suspensions

A previously described snRNA-seq procedure[66] was followed to increase the number of recovered cells from the human myometrium. After thawing and centrifuging cell samples, the supernatant was removed, and myometrial cell pellets were resuspended in 1 mL of lysis buffer on ice for 15 min. Samples were then transferred to a 15 mL conical tube and centrifuged at $500 \times g$ for 5 min at 4 °C. The resulting cell pellets were resuspended in 1× ST buffer within the 100–200 μL range, and nuclei solutions were passed through a 40 μm Falcon cell strainer. The concentration and viability of nuclei were evaluated using an EVE™ automated cell counter (NanoEnTek) with trypan blue. Finally, suspensions of ~10,000 single-nuclei were loaded onto Chromium Chips for the Chromium Single Cell 3′ Library preparation according to the manufacturer's recommendations (10x Genomics # PN-1000268).

## Single-cell capture, library preparation, and sequencing

To profile single cells/nuclei from three different areas of the human myometrium (anterior, posterior, and fundus), scRNA-seq analysis was performed using the 10X Chromium system (10X Genomics #1000204). Approximately 17,000 cells or nuclei were loaded onto a 10X G Chip to obtain Gel Bead-in-emulsions (GEMs) each containing an individual cell. GEMs were used to generate barcoded cDNA libraries following the manufacturer's protocol (Single Cell 3′ Reagent Kit v3.1, 10X Genomics, #PN-1000268) and quantified using the TapeStation High Sensitivity D5000 kit (Agilent, #5067-5593). Subsequently, gene expression libraries were constructed using 1–100 ng of each amplified cDNA library and quantified using the TapeStation High Sensitivity D1000 kit (Agilent, #5067-5584) to determine the average fragment size and library concentration. Libraries were normalized, diluted, and sequenced on the Illumina NovaSeq 6000 system (Illumina) according to the manufacturer's instructions.

## Spatial transcriptomics

The systematic mapping of cells and gene activity to tissue locations used the Visium Spatial Gene Expression Reagent Kit (10X Genomics, #PN-1000184). Eight full-thickness uterine samples were examined from eight women - three perimenopausal (under 55 years old) and five postmenopausal (equal to or older than 55 years old).

First, the quality of preserved RNA in FFPE blocks was evaluated based on the percentage of RNA fragments above 200 base pairs (DV200). Next, 7 μm-thick tissue sections were sampled using a semi-automated microtome (ThermoScientific HM340E). Per the manufacturer's protocol, each section was mounted onto a 6.5 × 6.5 mm capture area of the Visium Spatial Gene Expression Slide (10x Genomics #PN-2000233). Capture areas contain approximately 5000 barcoded spots, providing an average resolution of 1–10 cells (10X Genomics).

Tissue sections were then deparaffinized, H&E stained, and uncrosslinked according to the manufacturer's protocol (10X Genomics) with minor modifications. Brightfield images were taken using a 10X objective (Plan APO) on a Nikon Eclipse Ti2. Images were stitched together using NIS-Elements software (Nikon) and exported as .tiff files.

Following imaging and uncrosslinking, the Visium Spatial Gene Expression Slide & Reagent for FFPE kit (10XGenomics, #PN-1000184) was used for library construction. All steps (cDNA synthesis and amplification, library construction, and post-library construction quality control) were conducted according to the manufacturer's protocol. The libraries were sequenced on a HiSeqX (Illumina), 50PE (2× 150 bp), applying 1% Phix (Illumina, #FC-110-3001). Sequencing depth was calculated with the formula (Coverage Area × total spots on the Capture Area) × 50,000 read pairs/spot. Sequencing was performed using the following read protocol: read 1: 28 cycles; i7 index read: 10 cycles; i5 index read: 10 cycles; read 2: 90 cycles. Following

sequencing, data was visualized to determine each gene expression's spatial location and degree.

## sc/snRNA-seq data processing and filtering

Raw sequences were demultiplexed, aligned, and counted using the CellRanger software suite (v 6.0.2) for nuclei and whole cell gene expression calculations, which takes advantage of intronic reads to improve sensitivity and sequencing depth (human reference genome GRCh38-2020-A). Technical artifacts due to ambient RNA contamination were reduced with CellBender (0.2.0), and low-quality droplets and barcodes were filtered out in four quality control-based consecutive steps throughout the analysis: (i) low UMI-count barcode removal using an EmptyDrops-based method; (ii) cells/nuclei marked as doublets by DoubletFinder (2.0.3), using a custom pK value for each sample through the *find.pk* function provided in the package, and scds (1.6.0) tools – the hybrid approach from the scds R package was used to avoid removing false-positive doublets; (iii) cells/nuclei with median absolute deviation (MAD) > 3 in two of three basic quality control metrics: number of detected features, number of counts and mitochondrial ratio. These cell-to-count matrices were integrated and corrected using Seurat and Harmony functions, as described below. A final filtering step, (iv), was applied alongside different rounds of clustering, where the obtained clustered cells/nuclei with less than 750 features/cell, more than 25% mitochondrial ratio, and/or showing a pattern of high doublet-scoring plus no gene marker associated expression (during manual cell type annotations), were also removed.

Nuclei droplets were similarly processed, and quality controls were applied with the only specific criteria to remove nuclei with MAD > 2 in mitochondrial ratio per sample.

## Integration of single cells and single nuclei across conditions and clustering

As a first clustering analysis approach, read count matrices per sample were merged and processed following Seurat's default pipeline (package version 4.1.3). After normalization, the first thirty principal components on the 4000 highly variable genes were used for dimensional reduction; cells were clustered and projected onto the UMAP. *FindNeighbors* and *FindClusters* functions were then applied for graph-based clustering by constructing a KNN graph using Euclidean distance in the principal component analysis space, which was then defined into clusters using the Louvain algorithm to optimize the standard modularity function. clustree (R package v0.4.4) was applied to evaluate the impact of varying clustering resolution and select the optimal one as a balance of resolution and noise. The first output of sample distribution in clusters and cluster marker genes was then explored to evaluate biases from our data batches. Next, the Harmony R package (v1.0) was used to remove four primary sources of bias: i) between single-cell and single-nuclei protocols, ii) between tissue zone of origin (anterior, posterior, or fundus), iii) between aging conditions, and iv) the patient sample origin to remove inherent inter-individual differences. Dimensions were reduced as before, and an integration diversity penalty parameter (theta) of 2 was used. The top thirty Harmony components were used to embed and plot cells in the new reduced dimension space. These matrices were then input into Seurat's clustering and differential expression protocol. The clustering of different primary cell types for fine-grain cell type annotations was equally computed, following all the above-described steps.

## Tissue cell composition analysis and annotation of sc/snRNA-seq datasets

The identification and labeling of major cell types used the analysis of differentially expressed genes of each cluster compared to the remaining clusters using the Wilcoxon Rank Sum test and adjusted p-values for multiple comparisons with the false discovery rate (FDR) method. The primary cell populations were labeled by revising the

expression of reported canonical markers from each cluster. The five main cell types were then subset separately: endothelial cells, fibroblasts, SMCs, PV cells, and immune cells, and a new clustering was performed on each to create the different zoom-ins that describe the contained sub-populations. The clustered zoom-ins were manually annotated by an extensive review of differentially expressed genes in the Human Protein Atlas database, single-cell atlases, and scientific literature[13,17,67–77]. Genes with strong cluster-specificity, as determined by a p-value below 0.01, and the highest rank fold change and percentage of expressing cells were considered. Over-representation analysis on functional gene ontology and Reactome terms was also performed on cluster-specific genes with the WebGestaltR package (version 0.4.5) to help interpret cluster biological functions. Cell counts split by patient, cell type and menopausal state are provided in Supplementary Data 14.

## Analysis of differential cell abundances and gene expression between perimenopausal and postmenopausal myometria

Identifying cell subpopulations with differential abundance comparing perimenopausal and postmenopausal myometria as a model of aging used the Milo approach described by Dann et al.[78], which is available as a miloR package (version 1.6.0). This approach supports the grouping of cells on a k-nearest neighbor graph and evaluates the change in cell abundance between perimenopausal and postmenopausal conditions. Further differential gene expression analysis between conditions was performed using the Model-based Analysis of Single Cell Transcriptomics (MAST), where a contrast test was established for each cell type. The over-representation analysis on aging-associated genes identified enriched gene ontology biological processes and Reactome pathways.

## Spatial transcriptomics data processing

Spatial transcriptomics data processing used the Space Ranger v2.0.1 to align sequence data to the GRCh38-2020-A reference genome, followed by tissue and fiducial detection of spots and counting barcodes/unique molecular identifiers (UMIs) to generate feature-barcode matrices. These matrices were loaded into Seurat, and quality control filtering was performed based on the number of reads and detected genes per spot. Next, a reference matrix was built with informative genes from scRNA-Seq data (with a mean expression in a specific cell type at least 0.75 log fold higher than other cell types and removal of top 1% genes with the highest expression dispersion). Conditional autoregressive-based deconvolution was used to estimate cell composition in each spatial transcriptomics spot and build refined high-resolution maps of cell proportions and gene expression using the CARD package[79] (v 1.0). Specifically, the main cell populations were represented overlaid on the H&E image of the tissue, while the zoomed-in subpopulations were represented in "refined spatial maps". These refined spatial maps were high resolution spatial representations created by generating a regularly spaced grid within the tissue's outline, which was estimated using a 2D concave hull algorithm. This approach allowed for the estimation of new locations based on the conditional mean. Cell proportions estimated by CARD were normalized using the min-max method to validate differential abundance detected by scRNA-seq, while the spatial count matrix was processed using total count and log-normalization for validating differential expression detected by scRNA-seq. These data were then plotted in boxplots and contrasted differences between perimenopausal and postmenopausal myometrial samples by a Wilcoxon signed-rank test. Additionally, to evaluate spatial changes that may be associated with aging, we split the myometrial tissues into "proximal" and "distal" regions regarding their distance from the endometrium. These regions were defined by clustering spots at a low resolution (0.2) using Seurat's FindClusters function, so the cluster closest to the endometrium was labeled as "proximal" and the remaining (1 to 3 clusters) were labeled

as "distal". Next, CARD estimated cell type proportions were split by region (distal or proximal) and compared between peri and postmenopause by a Wilcoxon signed-rank test.

## Analysis of cell-cell communication

Identifying potential cell interactions between different cell populations in the aging human myometrium used the CellChat R package (v1.1.3)[16]. This tool infers the total interaction probability and communication information flows based on the expression of specific ligand-receptor pairs supported by a curated database. Briefly, the total interaction probability represents the probability of communication between two cell types, one acting as the sender and the other as the receiver, based on the number of interactor molecules expressed (ligand-receptor pairs) and the strength of this interaction (expression). Then, the sum of the communication probability of all pairs in a pathway network is used to calculate the communication information flow. For this analysis, pathways with less than ten cells were removed, and the influence of each cell population size was corrected by setting the population.size argument in the computeCommunProb function to TRUE. Lastly, differential CCC analysis between perimenopausal and postmenopausal myometria was performed with the ranknet function and a significance threshold of 0.05. For performing spatial-informed cell-to-cell communication analysis of the main cell types present in the tissue, spatial transcriptomics data was implemented using the CellChat library. Distance constraints to compute communication probabilities between cells were calculated with the function computeCommunProb(). Visualization of the communication results over the spatial layout were displayed by setting the "layout" parameter of the netVisual_aggregate() function to "spatial".

## Immunofluorescence of tissue sections

Myometrial tissue samples including a small fraction of endometrium were fixed in 4% paraformaldehyde and preserved in paraffin-embedded blocks. For immunostaining, tissue sections were deparaffinated and rehydrated. Antigen retrieval was performed with buffer citrate 1× at sub-boiling temperature for 10 min. Non-specific reactivity was blocked by incubation in 5% BSA/0.1% PBS-Tween 20 at room temperature for 30 min. Independent sections were incubated at 4 °C overnight with the following primary antibodies: 1:50 mouse monoclonal anti-human estrogen receptor 1 (ESR1) (Santa Cruz Biotechnology, catalog# sc-8002, clone# F-10, lot# G0908) and 1:50 rabbit monoclonal anti-human progesterone receptor (PGR), (Abcam, catalog# ab32085, clone# YR85, lot# GR237843-14). Moreover, additional sections were double stained for the detection of two proteins with the following primary antibodies: 1:50 mouse monoclonal anti-human alpha smooth muscle actin (αSMA) (Abcam, catalog# ab7817, clone# 1A4, lot# GR1009584-21) and 1:50 rabbit monoclonal anti-human voltage-dependent anion-selective channel proteins 1 and 2 (VDAC1/2) (Abcam, catalog# ab154856, clone# EPR10852(B), lot# GR219209-3 Then, slides were washed two times for 10 min with 0.1% PBS-Tween 20 before they were incubated for 1 h at room temperature with AlexaFluor-conjugated secondary antibodies (AlexaFluor 488 Abcam, catalog# ab150105, lot# GR3249866-4; AlexaFluor 594 Abcam catalog# ab150080, lot# GR3232361-2) diluted in 3% BSA/0.1% PBS-Tween 20 (1:1000). Finally, slides were washed two times in 0.1% PBS-Tween 20. To visualize nuclei, Pro-Long™ Diamond Antifade Mountant with 4′,6-diamidino-2-fenilindol (DAPI, ThermoFisher Scientific, #P36962) was utilized. Tissue sections were examined using a LEICA TCS-SP8 confocal microscope with a 40X objective. Finally, images were loaded in ImageJ[80] (v 1.53t) for nuclei segmentation and quantification of mean intensities per nuclei of nuclear proteins (ESR1 and PGR), which were then compared between peri and postmenopause with a Wilcoxon ranked sum test in R.

## Statistics & reproducibility

Statistical analyses were performed in software R (4.3.1) and no randomization was required since it was a descriptive study. Sample size was carefully predetermined, as conservative numbers towards the achievement of reproducible results. Samples that did not meet the minimum cell/nuclei count or failed to satisfy bioinformatic quality control thresholds were excluded from the analysis.

All samples within the same menopausal group (peri or postmenopause) were considered biological replicates. Thus, for the single-nuclei/single-cell experiments we had 6 biological replicates of perimenopause and 14 biological replicates of postmenopause. For spatial transcriptomics analysis we had 3 biological replicates of perimenopause and 5 biological replicates of postmenopause. All experiments included in the study were successful and no technical replicates were performed in this study. Sequencing runs were planned based on the availability of 16 single-nuclei/single-cell samples, without considering the primary variable (menopausal group).

## Reporting summary

Further information on research design is available in the Nature Portfolio Reporting Summary linked to this article.

## Data availability

Sequencing data (single-cell/nuclei RNAseq and spatial transcriptomics) supporting this study's findings have been deposited in the Gene Expression Omnibus (GEO) database under the accession identifier GSE236660. The uploaded data includes i) H5ad files containing the aggregated count matrices and metadata of each cell studied in the major cell populations and subpopulations; ii) Raw count matrices processed by 10X; iii) RNA ambient filtered count matrices processed by CellBender and iv) Images and filtered count matrix related to spatial scRNA-seq data. The raw sequences are not publicly available due to privacy concerns. However, they are available from the corresponding authors (A.M, amas@fundacioncarlossimon.com; C.S, carlos.simon@uv.es) upon reasonable request and with permission of the Institutional Review Board of the Spanish hospitals involved. Source data of box plots are also provided with this paper. Source data are provided with this paper.

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

## Acknowledgements

This study was jointly supported by the H2020-funded project Human Uterus Cell Atlas (HUTER 2020/2021) (Grant Agreement 874867), Miguel Servet Spanish Program Grant (CP19/00162), Health Research Funds (PI20/00942) from Carlos III Institute, Spain (AM), as well as Generalitat Valenciana (FDEGENT/2019/010) and (ACIF/2021/348) Ph.D. Training Grant for Valencian Entities (AML/PPJ). R.P. was supported by an Industrial Doctorate grant (DIN2020-011069) from the Spanish Ministry of Science and Innovation (MICINN). We acknowledge Javier Monleón, Beatriz Montero, Ana Ochando, Stuart P. Atkinson, and Carlos Lozano for contributing to this study. We would also like to thank Adrián González for his valuable help with the figure design.

## Author contributions

A.M. & C.S. designed the experiments, supervised the work and performed review and editing of the final manuscript. P.P.J., A.M.L., D.G., M.G.V., E.S.P., S.L.A., R.B., C.F.G. and performed the experiments. J.L.O., R.P.M., B.R., E.P.P., M.S. and A.M.L. performed the bioinformatic analysis. P.P.J. and A.M.L. wrote the original draft and contributed equally to this work. All authors have substantively revised the manuscript and approved the submitted version.

## Competing interests

The authors declare no competing interests.
