## [Peer Review File · Nature Communications]

Effect of Aging on the Human Myometrium at Single-Cell ResolutionREVIEWER COMMENTS

Reviewer #1 (Remarks to the Author):

This manuscript proposes to study age-related differences in uterine myometrium by comparing cell abundance, differential gene expression (DGE), pathways, cell-to-cell communications, and spatial localization of myometrial cell types and subtypes at single cell resolution in uteri from n=16 postmenopausal and n=4 peri-menopausal patients. Key findings herein are identification of 5 major myometrial cell types (endothelial, fibroblasts, smooth muscle (SMC), perivascular (PV), and immune, and 23 distinct subtypes among these) and differences in post- vs. peri-menopausal myometrium that include contractile cell abundance, DGE for endothelial and SMC contractility and voltage channels, fibroblast extracellular matrix-related genes, monocyte (mainly T-cell) inflammation-associated genes, altered and reduced cell-to-cell communications (CCC) and complete loss of CCC involving immune cell function, contractility, angiogenesis, tissue repair, and nervous system regulation. The concept of myometrial aging is important to understand dysfunctions of uterine homeostasis with age and, as the authors have suggested and clinical observations by others have demonstrated, age-related pregnancy complications due to myometrial dysfunction. Several items warrant author attention for clarity of data interpretation and rigor. These include:

1. Clinical metadata need more detail:
 - a. While Supplemental Table S1 provides some information, a more detailed table of clinical metadata is essential. Included should be individual subject listed (de-identified) and their corresponding age, BMI, whether cadaver-obtained or not, menstrual cycle phases of the pre-menopausal patients, and indication for surgery.
 - b. Note Line 403 says age range is 46-79, but Table S 1 says 46-83. Please clarify.
 - c. While the subject details section states patients had no gynecologic disorders, what specifically was excluded and how was this determined? It also states subjects had no malignancies (uterine or other?), or bacterial, fungal, or viral infections (at the time of obtaining the specimens or a history of such in the past? The latter is important as prior infection can alter myometrial features and function.
2. Was cadaver or non-cadaver status considered in the analysis? Was site of collection considered in the analysis? Did prior Cesarean section (myometrial trauma/repair) impact any of the data?
3. Some of the spatial localization panels (e.g., Fig 3D) have poor resolution, and the figure legends say “representative of” – of how many uteri?
4. As biospecimens were obtained from the uterine fundus and anterior and posterior walls, were there any differences in cell types, gene expression, other, among them? If not, please state so.
5. Does parity affect the data or could it? Would also be good in the Discussion to indicate something about this.
6. There is no mention of stem/progenitors in the peri - and post-menopausal myometrial samples. Please explain/clarify.
7. The observation of reduced cell-to-cell communications with aging is fascinating. Does this happen in other organs too – is it generalizable?
8. Line 304 – It is not strictly correct that the study uses the peri- to-post menopausal progression as a model system, as samples from the same subject going through the progression would be impossible. It is suggested that the authors indicate that the model focuses on the comparison of the post-menopause vs peri-menopause myometrium cell components.

9. Line 306 – others have performed scRNAseq on human myometrium mostly in the context of uterine fibroids (e.g., Goad Hum Reprod 2022; Tabula Sapiens Science 2022). It would be interesting to see if the data obtained therein can be compared to the current data if age range is compatible – either with the peri-menopausal or post-menopausal status of subjects in these studies. Even if not comparable, mention of others in this space would be appropriate.

10. Line 331 – should reference not only ref 7 but also Rosenthal BJOG 1998 – both of which show a gradual increase in obstetrical outcome as a continuous function. What about age range in your study?

11. The framework of the manuscript is myometrial dysfunction and poor pregnancy outcomes in older women. While the focus of data interpretation is solely on age, hormonal milieu is a key piece of peri- and post-menopausal physiology (which can be interrelated). Lack of information about steroid hormone receptors in the peri- and post-menopausal uteri and their cell types/ subtypes, in addition to no information about cycle phase in the four peri-menopausal subjects comprise a major gap in data analysis and interpretation. This should be included in the Results section, and in the Discussion in terms of biology and possible confounders.

12. Relevant to #11, poorer pregnancy outcomes with chronologic age can also be attributed to co-morbidities that increase with age – e.g., hypertension, diabetes, other. Including recognition of such in setting the premise for the study is important as otherwise the message could be interpreted as only age matters.

13. While natural pregnancy very rarely occurs in the menopause and egg or embryo donation occurs in some peri-menopausal women, embryo transfer is rarely done in recipients >54 years old. Thus, it may be confusing to the general reader why the main focus of the data interpretation is on the post-menopausal uterus (>54 years old). It is recommended that the authors consider their data in the context of a continuum of aging, as reference 7 and Rosenthal BJOG 1998 both suggest a steady rise in the process of myometrial aging leading to decline in function beginning in the 3rd decade of life (in a woman's 20's). Would be good to indicate the value of these studies with very focused populations eliminating other confounders as prior pregnancies, co-morbidities, etc. Also, data comparison of peri-menopausal myometrium with pre-menopausal myometrium in the same cycle phase would be more relevant to the aging myometrium and pregnancy outcomes. Can the authors expand the study to include such specimens?

14. Relevant to #12, besides pregnancy complications, what other dysfunctions of the post-menopausal myometrium may be impacted?

15. The observed diminished contractility of SMCs and endothelial and PV cells, disrupted immune responses, and altered interactions within/between cell types that impact inflammation, angiogenesis, fibrosis, and contractility may have relevance to poorer pregnancy outcomes in older pregnant women. Indeed, the review by Wu 2023 offers a comprehensive summary of morphologic changes (e.g., reduced myometrial mass, uterine fibrosis, blood vessel alterations), abnormal hormonal responses (not just to oxytocin), and blood vessel dysfunction. Comparing what has been done by others is important for the rigor of the data obtained herein and would significantly enrich the Discussion section and acknowledge others in this space.

16. The Discussion would do well to minimize repeating the results and expanding the data interpretation as described in #15 and to cell-type/subtype aging in other tissues, more broadly.

17. It is suggested at the end of the paper that strengths are listed and weaknesses expanded in the context of some of the comments above.

Reviewer #2 (Remarks to the Author):

Age-related myometrial dysfunction leads to various complications in women and thus it is of great significance to comprehensively understand the mechanisms of human myometrial aging. This study combines single-cell RNA sequencing and spatial transcriptomics to reveal the cellular and tissue structural changes associated with human myometrium aging. Overall, this study provides resource data to this field, but I do have some concerns regarding this manuscript as follows:

Major concerns:

1. Aging is characterized by progressive physiological changes, particularly functional decline throughout lifespan. I am wondering why the authors selected perimenopausal and postmenopausal women as samples to investigate aging. To better understand the age-associated molecular basis underlying human myometrium aging, the young samples should be included.
2. It is well known that the structure and function of the endometrium undergo dynamic changes during menstrual cycle. It is interesting to know whether and how age-related menstrual cycle irregularity contributes to human myometrium aging. The authors should clarify the detailed information for each donor and analyze the potential effects and mechanisms.
3. This manuscript is very descriptive without a focused point or pathway. The authors are suggested to perform an in-depth analysis into the bioinformatic data and mine the core regulatory mechanism underlying human myometrium aging. In addition, more experimental validations using alternative methods, such as immunostaining, functional assays by genetic manipulation, etc., are needed to elucidate the biological insight on human myometrium aging and improve the quality of this study. For example, reduced cell numbers and contractility-associated gene expression in the myometrial endothelial cell, immune-modulated or stressed changes in myofibroblast, lower contractile and ion-conductive activities of SMC, etc..
4. The spatial transcriptome data in this research only capture alterations in certain genes and pathways analyzed through single-cell/single-nucleus RNA-seq, and it is necessary to further investigate the spatial-specific changes during the aging process.
5. The author inferred cell-cell interactions only based on the expression of specific ligand-receptor pairs. To make the results more reasonable and reliable, an integrative analysis of single-cell transcriptome and spatial transcriptome data is required. Similarly, validation studies are suggested to provide biological insight on changes in cellular interactions.
6. Among the cell types identified, which cells are most sensitive to aging and are more likely to contribute to tissue aging.
7. It is a lack of details of the computational approaches and the provided code is poorly annotated and difficult to follow on the GitHub page.

Minor concerns:

1. What parameters for DoubletFinder were applied to the dataset? Were ground truths used or estimates? What pK values were used? What percentage of cells were found to be doublets?
2. How were batch effects corrected? Was a design matrix added during the scaling process? Were the effects of cell cycle genes regressed out?
3. What is resolution for clustering? A statistical approach to identify the resolution or find the number of clusters should be performed. For example, a sub-sampling with bootstrapping can be used. There are a variety of packages that can be used (i.e. see Patterson-Cross et al 2021 PMID: 33522897). Number of cells for each cell group and each sample should be clearly annotated.

Reviewer #3 (Remarks to the Author):

In this study, the authors present a single-cell atlas of the human myometrium in perimenopausal and postmenopausal women, illustrating the cell type-specific transcriptomic changes and alterations of cell-cell communication that accompany menopause. The authors included myometrial tissue samples from 6 perimenopausal and 14 postmenopausal women, and performed both single-cell and single-nucleus RNA-sequencing. Key findings of the study include the observation of fewer contractile capillary cells and diminished expression of genes related to ion channel expression in smooth muscle cells as well as a generalized impairment of gene expression across endothelial, smooth muscle, fibroblast, perivascular, and immune cells. Furthermore, altered myometrial cell-cell communication resulting in the loss of multiple signaling pathways was a hallmark of menopause. The authors conclude that these data can provide insight into the changes underlying complications faced by older women during pregnancy and labor.

Overall, the study represents a novel application of cutting-edge single-cell techniques to an increasingly relevant topic, namely how aging can affect cellular processes in the myometrium and subsequently complicate pregnancy in older women. However, the study is highly descriptive, and the authors may consider including additional analysis to expand on the functional changes in myometrial cells. Specific comments are as follows:

1. A key question arising from the authors' findings is how the observed transcriptomic changes translate to functional changes in individual cell types. If possible, the authors could consider performing functional determinations in myometrial SMCs or other target cell types and comparing these between perimenopausal and postmenopausal women.
2. Regardless of the feasibility of the abovementioned functional experiments, the authors should consider further mining their generated datasets to provide more insight into the functional alterations of each cell type with menopause. For example, additional application of GO, Reactome, or other annotated pathway analysis to the differentially regulated gene sets reported for each cell type could help elucidate which functions are altered, beyond the reporting of individual genes as done by the authors. Moreover, the application of the cell-cell communication analysis could also be expanded to show the individual interactions of each cell type (e.g., SMCs) with other cells, and whether the overall interactions are altered with menopause. Such additional analyses would be complementary to the consistent structure of each results section/figure focused on specific cell types. This strategy was previously reported (PMID: 35260533).
3. It would be interesting to see how the perimenopausal and postmenopausal myometrium compare to the myometrium during "healthy" reproductive ages (<45 years or so). Given that the collection of fresh samples could take some time, would it be feasible to leverage existing datasets for comparison?
4. Abstract: It is unclear why the authors specifically mention nervous system regulatory fibroblasts in the abstract, as this cell type is barely discussed in the rest of the manuscript. Consider removing to make room for more important findings.
5. Results/Figure 1: Did the authors note underrepresentation of larger cell types (e.g., SMCs) from the single-cell dataset? Other studies of the human reproductive tissues have noted some limitations of the microfluidics in terms of captured cell sizes (e.g., PMID: 35260533). Although the inclusion of single-nucleus sequencing would help overcome such limitations, it would still be worth mentioning any difficulties in this regard.
6. Results/Figure 1C: Did the authors attempt any overall comparative analysis between

spatial transcriptomic maps of the myometrium? It would be interesting to note if trends for differing localization of major cell clusters could be observed post-menopause, as this could partly relate to the reported changes in cell-cell communication.

7. Results/Figure 1D: The term “information flow” should be explained in the results section, as this may not be immediately apparent to readers.

8. Results/Supplementary Figure 3: The spatial transcriptomics should include quantification, as shown in other figures, to support the specific cell types or genes mentioned.

9. Results/Figure 5: What are the characteristics of the “undifferentiated” T cells? Is it possible that this subset includes innate lymphoid cells or MAIT cells?

10. Figure 1: For Figure 1B, please move the color legends to be close to the top of the figure panel, to aid in interpreting each color for the pie charts and cell types. In addition, there is unnecessary white space between Figure 1B and 1D that could be reduced to improve the “squareness” of the figure.

11. Figure 2: For the plot types used in Figure 2B-C (similar plots are in other figures as well) it is not immediately apparent how the colors and changes relate to the study groups (i.e., what is increased/decreased with menopause). Some additional labelling or other modifications would be helpful. In addition, the cluster names in Figure 2D should be larger, as these are currently too similar to other less important plot labels.

12. Line 288 has a typo, the figure citation here is likely meant to be Fig. 7E.

We would like to express our gratitude to the three reviewers for their valuable insights, which have enabled us to enhance and refine our work. We have addressed all their comments and questions in our point-by-point document below with additional experiments, data, and detailed responses.

Reviewer#1 Page 1-10

Reviewer#2 Page10-16

Reviewer#3 Page 16-21

We have included a concise index at the end of this document outlining all the updated figures, tables, and new data (page 21-22). Additionally, we have provided a compilation of references used throughout this point-by-point reply to the reviewers (page 22-24).

REVIEWER COMMENTS

Reviewer #1 (Remarks to the Author):

This manuscript proposes to study age-related differences in uterine myometrium by comparing cell abundance, differential gene expression (DGE), pathways, cell-to-cell communications, and spatial localization of myometrial cell types and subtypes at single cell resolution in uteri from n=16 postmenopausal and n=4 peri-menopausal patients. Key findings herein are identification of 5 major myometrial cell types (endothelial, fibroblasts, smooth muscle (SMC), perivascular (PV), and immune, and 23 distinct subtypes among these) and differences in post- vs. peri-menopausal myometrium that include contractile cell abundance, DGE for endothelial and SMC contractility and voltage channels, fibroblast extracellular matrix-related genes, monocyte (mainly T-cell) inflammation-associated genes, altered and reduced cell-to-cell communications (CCC) and complete loss of CCC involving immune cell function, contractility, angiogenesis, tissue repair, and nervous system regulation. The concept of myometrial aging is important to understand dysfunctions of uterine homeostasis with age and, as the authors have suggested and clinical observations by others have demonstrated, age-related pregnancy complications due to myometrial dysfunction.

We thank Reviewer #1 for the insight and remarks about age-related reproductive and obstetric complications due to myometrial dysfunction. We have made efforts to incorporate and reflect upon all the valuable input provided.

Several items warrant author attention for clarity of data interpretation and rigor. These include:

1. Clinical metadata need more detail:

a. While Supplemental Table S1 provides some information, a more detailed table of clinical metadata is essential. Included should be individual subject listed (de-identified) and their corresponding age, BMI, whether cadaver-obtained or not, menstrual cycle phases of the pre-menopausal patients, and indication for surgery.

A1a: As requested by the reviewer, we have provided a more detailed table of clinical information containing metadata for individual subjects such as age, BMI, menopausal status, and indication of surgery among others (see New Supplementary Table 1).

Regarding the phases of the menstrual cycle, it is important to clarify that the individuals in our study were in the peri-menopausal and post-menopausal stage, not the pre-menopausal stage. This distinction is significant because, as outlined in the methods section (page 17, lines 513-516), the presence of hormonal fluctuations and unpredictable menstrual cycle patterns in the perimenopause and absent in the postmenopause makes it challenging for us to pinpoint a specific menstrual phase. Further limitations emerged because of the inclusion of patients from an organ donor program, and regrettably, we couldn't gather these details from their family members.

While information related to cadaver has been included as “uterine removal from organ donor programme”, we'd also want to emphasize that the myometrial samples were acquired through the surgical extraction of the entire uterus after meeting the criteria for brain death, following the protocol approved by the ethics committee. This approach permits the recovery of organs before circulatory arrest, thus minimizing the duration of warm ischemia. The surgical removal of the uterus was executed using a method of static cold preservation, in the same planned surgery for the removal of the organs to be transplanted. Subsequently, the uterus was submerged in a preservation solution and transported within a temperature range of 3-5 °C. This procedure mirrors the process for uterus transplant, which indeed is the same we followed for uteri obtained from hysterectomies conducted due to prolapse. To make this clear, authors have developed this point in page 18, lines 527-536.

b. Note Line 403 says age range is 46-79, but Table S 1 says 46-83. Please clarify.

A1b: Your vigilance in catching this error is greatly appreciated. There was, in fact, a typographical error in Table S1. We have rectified it in the new Table S1, which now accurately displays the updated age range as 46-79.

c. While the subject details section states patients had no gynaecologic disorders, what specifically was excluded and how was this determined? It also states subjects had no malignancies (uterine or other?), or bacterial, fungal, or viral infections (at the time of obtaining the specimens or a history of such in the past? The latter is important as prior infection can alter myometrial features and function.

A1c: Thank you for rising this issue. A more detailed description of exclusion criteria has been added in page 18, lines 537-544. Specifically, we excluded individuals from the study if they had gynaecological conditions such as endometriosis, uterine malformations, uterine leiomyoma, endometrial polyp/hyperplasia, uterine septum, Asherman's syndrome, or hydrosalpinx, as confirmed by the Pathological Anatomy services at the data collection site. Patients were not considered for inclusion in the study if they had previously been diagnosed with malignancies unrelated to the uterus and if such malignancies were also identified during the surgical procedure. To mitigate the potential effects of previous infections, inclusion in the study also necessitated negative serological tests for HIV, HBV, HCV, and RPR.

2. Was cadaver or non-cadaver status considered in the analysis? Was site of collection considered in the analysis? Did prior Cesarean section (myometrial trauma/repair) impact any of the data?

A2: Authors appreciate your diligence in highlighting these concerns regarding the various clinical variables inherent to our analyses. To evaluate whether the cells exhibit any unreported distribution associated to these clinical variables, we have generated new UMAP and bar plot representations considering each factor of interest independently (included now as Supplementary Figure 2). All the factors showed a uniform distribution of cells across all reported clusters, demonstrating that none of them imprint different expression patterns in any cell population. We also wanted to emphasize the subsequent aspects:

- Although we have already addressed in #A1a the distinction between cadaver (brain death) and non-cadaver status (living), indicating that the procedure for obtaining uteri from organ donors closely resembles that of uteri obtained from hysterectomies due to prolapse, we have conducted the comparison of conditions to identify any potential pattern that could influence the analyses. As illustrated in Supplementary Figure 2C and D, the results reveal a uniform presence of cells in both brain death and living individuals in all identified cell clusters, demonstrating that this status do not imprint trends in any cell type.
- In terms of the collection site, we also conducted comparative analyses across the three hospitals (Hospital General Universitario Valencia, Hospital Clinic Universitario Valencia, and Hospital La Fe Valencia) where the samples were obtained. As described in results section (pag 4, lines 88-91) and demonstrated in Supplementary Figure 2A and B, we observed no significant differences between the various collection sites, indicating that samples from all three hospitals are comparable to each other and do not introduce any variability to report into the analyses.
- Lastly, with respect to the data on cesarean sections, Table S1 has been revised to include related information for those patients for whom it was available in their clinical records. In brief, only one patient underwent cesarean section surgery prior to childbirth, four patients had an "unknown" cesarean section status, while the remaining patients had not undergone cesarean section surgery. Given this data, conducting comparative analyses between these groups is not feasible because the number of patients who underwent cesarean section is not comparable to the number of patients who did not. Nonetheless, we conducted comparative analyses to assess if parity could affect the analysis. As shown in Supplementary Figure 2E and F, the results indicate a consistent presence of cells in nulliparous (0) and multiparous (1, 2, 3, 4, 5) individuals.

3. Some of the spatial localization panels (e.g., Fig 3D) have poor resolution, and the figure legends say “representative of” – of how many uteri?

A.3: We appreciate your comment and acknowledge the confusion because we did not provide a detailed explanation. The spatial location panels in Figures 2E, 3D, 4E and S3B (old version) are “refined spatial maps”. This means that the original tissue spots have been subdivided into a grid of smaller positions, allowing for the estimation of cell type compositions and expression at a significantly greater spatial resolution than what was initially measured in the original study (Ma et al., 2022). The images in the figures depict these grids, which is why they might seem "pixelated," but it's essential to note that they are at the highest resolution possible, which is 600 dots per inch (dpi). This specific information has been included in the “Methods section” (page 24, lines 707-712).

Regarding the figure legends, when we used the term "representative of," we are referring to samples that were part of the total samples analysed using spatial transcriptomics. As detailed in the Methods section, eight full-thickness uterine samples were examined in total, consisting of three peri-menopausal and five post-menopausal samples. To ensure clarity, we have explicitly mentioned in the figure legends.

4. As biospecimens were obtained from the uterine fundus and anterior and posterior walls, were there any differences in cell types, gene expression, other, among them? If not, please state so.

A4: We have assessed our dataset while considering the uterine region. Regarding a potential unaccounted source of variation, the dataset shows that the uterine zones are uniformly spread, and no apparent biases have been identified during the exploratory analysis (Supplementary Fig.1C). Moreover, when we conducted a differential abundance analysis to investigate possible variations in cell abundance across the uterine zones (ZP: zone posterior; ZA: zone anterior; ZF: fundus) in the context of menopause (Post: postmenopause; Peri: perimenopause), we did not observe any statistically significant results (see figure below).

Finally, the pseudobulk differential expression analysis carried out with DESeq2 showed no significant differences in gene expression between tissue regions.

5. Does parity affect the data or could it? Would also be good in the Discussion to indicate.

A5: After following the analysis outlined in #A2, we assessed how parity could impact the data. In line with the remaining clinical variables, there were no distinctive patterns identified in parity, as shown in Supplementary Figure 2E and F.

6. There is no mention of stem/progenitors in the peri - and post-menopausal myometrial samples. Please explain/clarify.

A6: We appreciate this comment and acknowledge the relevance of stem/progenitor cells in myometrial functionality and homeostasis with aging. Unfortunately, we were unable to identify any cell populations exhibiting a stem cell profile using the markers that have been previously described in human myometrium, including ABCG2 (Ono et al., 2007), CD44/STRO1 (Mas et al., 2015), CD34/CD49f (Ono et al., 2015), and SUSD2 (Paul et al., 2023). Additionally, we did not find cycling cells, which would be expected for stem/progenitor cells. In our opinion, this is because stem/progenitor cells are present at very low proportions within adult human tissues (Bhartiya, 2021). We have added a concise paragraph on page 15, lines 444-446, to underscore this limitation.

7. The observation of reduced cell-to-cell communications with aging is fascinating. Does this happen in other organs too – is it generalizable?

A.7: Thank you for this appreciation. As discussed by López-Otín et al., 2023, altered cell communication is considered a hallmark of aging, which significantly impacts homeostatic regulation in the complete body. While previous research has linked aging-related impairments in angiogenesis, immune function, and tissue repair to a decline in 'beneficial' intercellular communication and an increase in 'detrimental' intercellular communication in various organs (Fafián-Labora et al., 2020), our work represents one of first studies highlighting a decrease/loss of cell communication as a key contributing factor associated with myometrial aging. We have included a brief paragraph in page 13, lines 370-374, to emphasize this observation.

8. Line 304 – It is not strictly correct that the study uses the peri- to-post menopausal progression as a model system, as samples from the same subject going through the progression would be impossible. It is suggested that the authors indicate that the model focuses on the comparison of the post-menopause vs peri-menopause myometrium cell components.

A8: We agree with your comment. To enhance the accuracy of our work's description, we have made an adjustment to page 12, line 330, which now reads as: “a perimenopause versus post-menopause comparison as a model system”.

9. Line 306 – others have performed scRNAseq on human myometrium mostly in the context of uterine fibroids (e.g., Goad Hum Reprod 2022; Tabula Sapiens Science 2022). It would be interesting to see if the data obtained therein can be compared to the current data if age range is compatible – either with the peri-menopausal or post-menopausal status of subjects in these studies. Even if not comparable, mention of others in this space would be appropriate.

A.9: Thank you for your comment. Certainly, it is essential to acknowledge the significant contributions made in the field of single-cell analysis of the human myometrium (page 12, lines 328-330 and references section). However, it should be noted that the works indicated differ from our study in several key aspects. Firstly, the study conducted by Goad et al. in 2022 primarily focuses on myometrium with fibroids, which is distinct from our dataset consisting

of unaffected myometria by gynaecological issues. Secondly, the research conducted by Tabula Sapiens, while undertaking a comprehensive exploration of human organs and tissues, approaches the uterus as a whole entity without distinguishing between endometrial and/or myometrial tissues. Consequently, it does not provide specific insights into the cellular populations originating from the aging myometrium.

10. Line 331 – should reference not only ref 7 but also Rosenthal BJOG 1998 – both of which show a gradual increase in obstetrical outcome as a continuous function. What about age range in your study?

A.10: We acknowledge reviewer's concerns about the age range used on this study. First, we have updated the reference list, including in addition to those already referenced, the recommended paper by Rosenthal and Peterson-Brown, 1998. Besides, a minor adjustment has been made to the introduction (page 3, lines 64-67) to incorporate this publication into our research as well.

Regarding the age range that the reviewer is asking us about, we have used two age groups: perimenopausal (ranging from 46-54 years old) and postmenopausal (from 54 years old onwards) due to the following reasons:

- 1) Our study includes a small sample size (n=20) when compared to the studies by Rosenthal & Peterson-Brown (n=6410) and Main et al., 2000 (n=8498). Because of the gaps of sample coverage along age in our data, the patients' samples will not be well fitted to the age as a continuous variable of time. We should note here that methodological studies on this matter (Spies et al 2019) report higher statistical power in the classical pairwise comparison approach on short time series for RNAseq experiments rather than those of more complex time course approaches. Furthermore, the limitation in sample collection arises from the difficulty in obtaining myometrial samples non-invasively, resulting in most patients being included in advanced age groups since they are from pelvic prolapse.
- 2) We examined cellular and molecular changes in the tissue that may arise due to the aging process, being related, or not, to obstetric outcomes, whereas in the studies by Rosenthal & Peterson-Brown, 1998 and Main et al., 2000, age is directly linked to the obstetric outcome itself, making them non-invasive observational studies.

The fact that both studies (Rosenthal&Peterson-Brown,1998;& Main et al., 2000) show that obstetric outcomes are a continuous function of age and begin to be affected from early ages in a woman's reproductive life, is not an excluding condition for these results to be also observed when comparing two specific age groups, as is the case in our study. As a clarification, we have mentioned these limitations in the discussion (page 16, lines 459 to 465).

11. The framework of the manuscript is myometrial dysfunction and poor pregnancy outcomes in older women. While the focus of data interpretation is solely on age, hormonal milieu is a key piece of peri- and post-menopausal physiology (which can be interrelated). Lack of information about steroid hormone receptors in the peri- and post-menopausal uteri and their cell types/ subtypes, in addition to no information about cycle

phase in the four peri-menopausal subjects comprise a major gap in data analysis and interpretation. This should be included in the Results section, and in the Discussion in terms of biology and possible confounders.

A.11: Thank you for this comment. While perimenopause women display unpredictable patterns and hormonal fluctuations that make not possible to discern among different menstrual phases, we concurred that the hormonal environment plays a pivotal role in the physiology during peri- and post-menopause. Hence, to assess the potential hormonal alterations occurring in the myometrium because of aging, we further investigated the expression of estrogen (ESR) and progesterone (PGR) receptors, as shown in Results section (page 10, lines 276-285) and Supplementary Figure 7.

Despite the absence of notable differences in ESR2 expression, ESR1 showed an increased expression in all cell subpopulations during post-menopause, a finding that was additionally confirmed through both mRNA and protein analysis. Conversely, PGR exhibited a decrease in expression exclusively in contractile SMC during postmenopause, as indicated by the differential expression findings, and further confirmed through spatial transcriptomics and immunofluorescence.

ESR1 has been implicated in several aging-related processes, such as vascular health (Davezac et al., 2021), and it also plays a pivotal role in maintaining myometrial quiescence during pregnancy (Anamthakula et al., 2019), emphasizing its significance in myometrial physiology. As part of the discussion (page 14, lines 419-429), we hypothesize that myometrial cells could be upregulating the expression of estrogen receptors to maximize their sensitivity to estrogen, as a compensatory mechanism in response to the reduction of estrogen levels associated with aging.

While PGR is a well-established regulator of myometrial contractility, at the single-cell level, we are unable to differentiate whether this downregulation pertains to isoform A or B. As Peavey et al., reported in 2021, these isoforms have completely opposed functions, with one promoting a contractile state and the other facilitating relaxation. To address this ambiguity, we conducted immunofluorescence using an isoform B-specific antibody, which confirmed the downregulation of this isoform. These findings provide further support for the altered contractility being a key feature of myometrial aging.

12. Relevant to #11, poorer pregnancy outcomes with chronologic age can also be attributed to co-morbidities that increase with age – e.g., hypertension, diabetes, other. Including recognition of such in setting the premise for the study is important as otherwise the message could be interpreted as only age matters.

A.12: We appreciate your comment. We share your perspective on the importance of acknowledging this potential limitation, and we recognize that specific sections of the text may not accurately convey this. Consequently, we have implemented several modifications in the text:

-We have adjusted line 26 in the abstract to prevent any misunderstanding that myometrial dysfunction is the sole cause.

-We've inserted a sentence at page 3, lines 62-63 within the introduction to elucidate that the increase in maternal mortality with advancing age may be attributed to uterine conditions or other age-associated factors like hypertension or diabetes.

- We've included a sentence at page 16, line 468-473 to acknowledge that this study's scope is limited to the myometrium. However, it emphasizes the need for a comprehensive approach that considers various uterine tissues, other organs, and age-related conditions (such as hypertension, diabetes, and cancer) to gain a holistic understanding of the intricate mechanisms involved in poorer pregnancy outcomes associated with advancing maternal age.

13. While natural pregnancy very rarely occurs in the menopause and egg or embryo donation occurs in some peri-menopausal women, embryo transfer is rarely done in recipients >54 years old. Thus, it may be confusing to the general reader why the main focus of the data interpretation is on the post-menopausal uterus (>54 years old). It is recommended that the authors consider their data in the context of a continuum of aging, as reference 7 and Rosenthal BJOG 1998 both suggest a steady rise in the process of myometrial aging leading to decline in function beginning in the 3rd decade of life (in a woman's 20's). Would be good to indicate the value of these studies with very focused populations eliminating other confounders as prior pregnancies, co-morbidities, etc. Also, data comparison of peri-menopausal myometrium with pre-menopausal myometrium in the same cycle phase would be more relevant to the aging myometrium and pregnancy outcomes. Can the authors expand the study to include such specimens?

A13: Thank you for pointing your concern about the age ranges and your suggestion to consider age as a continuous variable. In response to your suggestion, we have emphasized the importance of such research by making comparisons to our own study (page 16, lines 459-465).

As we have indicated in answer A.10, our sample size at single cell resolution is not sufficiently large to establish more than two age groups. Consequently, the observed changes may indeed be related to age, but we are unable to determine whether they manifest suddenly at any specific time point or are the result of a continuous process.

It's worth noting that the patients included in our study may have additional unknown underlying health conditions that could act as confounding factors but this is out of our perimeter.

Unfortunately, and related with the inquires raised in question #10, it is not feasible to expand the study including healthy pre-menopausal individuals because myometrial analysis requires the complete organ, unless they come from the organ donation program. However, even in such instances, acquiring organs for research purposes presents significant challenges. As a result, our sample pool is exclusively composed of patients who have undergone uterine prolapse procedures or are part of the national transplant program. This circumstance renders it unfeasible for us to acquire samples from individuals outside of these specific contexts, thus impeding our ability to incorporate younger patients into the study.

Although nowadays embryo transfer is rarely done in recipients >54 years old, this trend is increasing as is the women's life expectancy in the first world. Precisely, this population has

the higher risk of obstetrical complications being the objective of our basic study.

14. Relevant to #12, besides pregnancy complications, what other dysfunctions of the post-menopausal myometrium may be impacted?

A.14: We share your perspective on the importance of recognizing conditions that can arise from the aging of myometrial tissue, not limited to pregnancy-related complications. We have provided a brief mention of some of these characteristics in the introduction (page 3, lines 64-67). Furthermore, we have incorporated an additional paragraph within the discussion (page 17, lines 489-495) addressing potential dysfunctions that may arise, particularly in the context of labor initiation. These encompass dystocia, uterine atony, preterm labor, and postpartum hemorrhage.

15. The observed diminished contractility of SMCs and endothelial and PV cells, disrupted immune responses, and altered interactions within/between cell types that impact inflammation, angiogenesis, fibrosis, and contractility may have relevance to poorer pregnancy outcomes in older pregnant women. Indeed, the review by Wu 2023 offers a comprehensive summary of morphologic changes (e.g., reduced myometrial mass, uterine fibrosis, blood vessel alterations), abnormal hormonal responses (not just to oxytocin), and blood vessel dysfunction. Comparing what has been done by others is important for the rigor of the data obtained herein and would significantly enrich the Discussion section and acknowledge others in this space.

A.15: We truly appreciate reviewer's input in this regard. Indeed, incorporating references to the work of others enhances the credibility of the presented results and allows for better comparisons with previously published research. In fact, for our article, we had also relied on the publication by Wu et al., 2023. Following the reviewer's suggestion, we have enriched the discussion with the work of other authors, thus making it more robust and rigorous by incorporating contributions from other authors, thereby enhancing its robustness and rigor. Consequently, we have added several comments and references with other works supporting our results in relation to defective senescence (page 13, lines 362-365), angiogenesis (page 13, lines 384-385) and collagen-deposition (page 13, lines 388-389).

16. The Discussion would do well to minimize repeating the results and expanding the data interpretation as described in #15 and to cell-type/subtype aging in other tissues, more broadly.

A.16: We agree with the comment raised by the reviewer, and we have expanded the data interpretation throughout the Discussion section.

17. It is suggested at the end of the paper that strengths are listed, and weaknesses expanded in the context of some of the comments above.

A.17: In response to the suggestions provided by the reviewer, we have extended the discussion by including:

- List of weaknesses:
 - We have enhanced the section discussing the potential bias stemming from cell size and the possible underrepresentation of larger and minor cell types, such as SMCs, in the dataset (page 15, lines 440-444).
 - We've included a paragraph that addresses the choice of age groups rather than treating age as a continuous variable, which is in response to the reviewer's inquiries raised in questions #10 and #13 (page 16, lines 459-465).
 - We have included a paragraph (page 16, lines 468-473) to emphasize that the aging process is not solely dependent on myometrial conditions but is intricately linked to the interactions of various physiological processes within the body. This addition addresses the reviewer's question #12.
- List of strengths:
 - The number of single-cells / single-nuclei collected for this work is notably larger in comparison to previously published atlases focused on the myometrium. Detailed explanation can be found in page 16, lines 477-481.
 - The combination of two innovative methods such as sc/snRNAseq plus sc-transcriptomics enhances the quality and robustness of the presented data. This statement has been added in page 16, lines 481-484.

Reviewer #2 (Remarks to the Author):

Age-related myometrial dysfunction leads to various complications in women and thus it is of great significance to comprehensively understand the mechanisms of human myometrial aging. This study combines single-cell RNA sequencing and spatial transcriptomics to reveal the cellular and tissue structural changes associated with human myometrium aging.

We would like to express our gratitude to Reviewer #2 for their insightful comments about our manuscript.

Overall, this study provides resource data to this field, but I do have some concerns regarding this manuscript as follows:

Major concerns:

1. Aging is characterized by progressive physiological changes, particularly functional decline throughout lifespan. I am wondering why the authors selected perimenopausal and postmenopausal women as samples to investigate aging. To better understand the age-associated molecular basis underlying human myometrium aging, the young samples should be included.

A1: The authors are in complete alignment with the approach recommended by the reviewer. In fact, an ideal scenario would encompass the inclusion of younger patients to showcase the myometrium's functional decline throughout lifespan. Unfortunately, it is not feasible to expand the study including healthy younger individuals. The limitation arises from the inability to obtain myometrial samples in younger patients without surgical indications, mainly due to medical, ethical, and legal constraints because the whole organ should be extracted to have access to the whole myometrial layer. Consequently, our sample group comprises exclusively of patients who have undergone uterine prolapse procedures or were enrolled in the national

transplant donor program. This situation makes it impractical for us to obtain samples from individuals not falling within these contexts, thereby hindering our capacity to include younger patients in the study.

It's worth noting that we have made efforts to bridge the data gap from younger individuals by consulting other myometrium atlases that have been previously published. First, the study conducted by Goad et al. in 2022 primarily concentrates on myometrium with fibroids, whereas our dataset comprises unaffected myometrial tissue without gynecological issues. Secondly, the research conducted by Tabula Sapiens in 2022, while comprehensively exploring human organs and tissues, treats the uterus as a unified entity without making distinctions between endometrial and myometrial tissues. Lastly, the investigation by Pique-Regi et al. in 2022 involves younger patients, yet the data isn't directly comparable since it pertains to myometria during parturition, which represents a physiological process significantly distinct from "steady-state myometria." Consequently, it does not provide specific insights into the cellular populations originating from the aging myometrium.

We have incorporated appropriate justifications and explanations for these limitations (page 16, lines 465-468).

2. It is well known that the structural and function of the endometrium undergo dynamic changes during menstrual cycle. It is interested to known whether and how age-related menstrual cycle irregularation contributes to human myometrium aging. The authors should clarify the detailed information for each donor and analyze the potential effects and mechanisms.

A2: Thank you for bringing your concern about the effect of menstrual cycle to human myometrium aging. On this regard, we have provided a more detailed table containing all clinical information we have been able to recapitulate from hospitals, including menopausal status among others. Unfortunately, analysing the influence of menstrual phases presents a challenge in this study, since the individuals included are in the peri-menopausal stage rather than being pre-menopausal. As detailed in the methods section (page 17, lines 516-518), the peri-menopausal stage is featured by unpredictable menstrual cycle patterns and hormonal fluctuations, that are absent in the postmenopause making it difficult for the study to pinpoint a specific menstrual phase. Additionally, the inclusion of patients from the organ donor program complicates the ability to gather menstrual cycle details from their family members.

3. This manuscript is very descriptive without a focused point or pathway. The authors are suggested to perform an in-depth analysis into the bioinformatic data and mine the core regulatory mechanism underlying human myometrium aging. In addition, more experimental validations using alternative methods, such as immunostaining, functional assays by genetic manipulation, etc., are needed to elucidate the biological insight on human myometrium aging and improve the quality of this study. For example, reduced cell numbers and contractility-associated gene expression in the myometrial endothelial cell, immune-modulated or stressed changes in myofibroblast, lower contractile and ion-conductive activities of SMC, etc.

A3: We thank the author for this suggestion. To gain a deeper understanding of our data, we have conducted a comprehensive analysis using several bioinformatics approaches and experimental/in-silico validations as listed below:

- 1) We have demonstrated through differential abundance and differential expression analysis that the uterine zones were uniformly distributed, and no significant differences were found in the human myometrium context of menopause.
- 2) In an attempt to showcase the myometrium's functional decline throughout lifespan, we have made efforts to bridge the data gap from younger individuals by consulting other myometrium atlases that have been previously published (Goad et al., 2022; Tabula Sapiens, 2022; Pique-Regi et al., 2022), but unfortunately, they do not provide specific insights into the cellular populations originating from the aging myometrium.
- 3) To gain a better understanding of spatial changes that may happen within the aging myometrium, we have assessed trends in the localization of major cell clusters by comparing the distal and proximal myometrial regions by taking the endometrium as a reference.
- 4) To provide further biological insight on changes in cellular interactions, an integrative analysis of cell-to-cell communication and spatial transcriptome data has been performed. Notably, as shown in Supplementary Figure 6, the tissue-level validation of various signalling networks (specifically ANGPTL, CXCL, and PDGF) revealed intriguing instances of cell-to-cell communication within major cell types (including SMC, fibroblasts, VSMC, and endothelium), allowing us to ascertain both the direction and intensity of these interactions.
- 5) While our primary focus of data interpretation centers on aging myometrium, it's essential to note that the hormonal landscape is a critical factor to consider between peri- and post-menopausal women physiology. In this sense, we have explored the expression of oestrogen and progesterone receptors (ESR1, ESR2 and PGR) at mRNA (scRNA-seq and spatial transcriptomics) and protein level (immunostaining), as shown in Supplementary Figure 7, and supporting findings have been presented in results section (in page 10, lines 276-284) and in discussion (page 14 lines 419-429).
- 6) After establishing the increased abundance of smooth muscle cells in the aging myometrium, we corroborated the decline in their contractile function associated with aging through various means, including differential gene expression (Fig. 4D), spatial transcriptomics (Fig. 4E; Supplementary Fig. 4D and E), and immunostaining (Fig. 4F).
- 7) Finally, to investigate the core processes underlying the aging process in the myometrium, we evaluated which cell populations are most affected in our dataset. This has specifically pinpointed stimuli-responsive SMC, canonical SMC, immune-modulated fibroblasts, and stimuli-responsive VSCM as the most affected cell types, once again highlighting the impairment of contractility, immune response, fibrosis and angiogenesis that happens in the aging myometrium.

Consequently, considering all the evidence we have presented, we trust that we have satisfactorily addressed the reviewer's concerns.

4. The spatial transcriptome data in this research only capture alterations in certain genes and pathways analysed through single-cell/single-nucleus RNA-seq, and it is necessary to further investigate the spatial-specific changes during the aging process.

A4: We appreciate your comment and we have made efforts to maximize the utilization of the findings from our spatial analysis. However, it's essential to acknowledge that spatial transcriptomics is an emerging field, and specific tools for comparing various conditions have yet to be developed. To investigate spatial-specific changes during the aging process, we have conducted a comparative analysis of changes in cell proportions between the distal and proximal regions of the myometrium in relation to the endometrium. To incorporate these new results, we have implemented the following changes in the text:

- Methods section provides a detailed explanation of this analytical approach (page 24, lines 718-725).
- Supplementary Figure 4 includes a new boxplot comparing tissue regions and peri or postmenopausal status. This has been also referenced in the Results section (page 7, lines 189-191).
- Discussion also includes possible explanations for the observed regional differences (page 12, line 356-360), and provides a potential rationale for the disparities between scRNAseq and spatial transcriptomics (page 15, lines 449-457).

Furthermore, we have integrated spatial transcriptomics data with CCC data from the scRNAseq analysis, which is now included in the manuscript (page 10, lines 269-272).

5. The author inferred cell-cell interactions only based on the expression of specific ligand-receptor pairs. To make the results more reasonable and reliable, an integrative analysis of single-cell transcriptome and spatial transcriptome data is required. Similarly, validation studies are suggested to provide biological insight on changes in cellular interactions.

A5: We utilized single-cell transcriptomics data to deconvolve the spatial transcriptomics data, effectively integrating these two distinct technologies. However, to extend our insights beyond these findings, we conducted additional analyses that take into account spatial-informed cell-cell interactions.

Given the limited resolution of spatial transcriptomics data, we focused on major cell types (SMC, fibroblasts, endothelial cells and VSMC) for this integration. Interestingly, we have been able to find CCC interactions among different cell types located at the spatial level. This new approach has been included in the manuscript as follows:

- The analysis used has been described in the methodology section (page 25, lines 740-745).
- Supplemental Figure 6 has been added to the manuscript, as well as related results for this CCC have been added to the manuscript in page 10, lines 269-272.

However, it's crucial to approach these results with caution as they may exhibit slight discrepancies when compared to those derived from sc/sn RNAseq. This discrepancy arises from the limitation of the deconvolution process using sc/sn transcriptomics data, as the spatial slide only displays the most representative cell types for each spot. In fact, we have

acknowledged this limitation in the discussion section (page 15, lines 449-457).

6. Among the cell types identified, which cells are most sensitive to aging and are more likely to contribute to tissue aging.

A6: Excellent point. After delving into the DE results as well as the cell-to-cell communication signalling networks, we observed several populations more susceptible to the aging process, as they display the highest number of DE genes between peri and postmenopause, as well as notable changes in cell-to-cell communications.

When examining specific subpopulations, we found SMC (stimuli-responsive SMC, contractile SMC and canonical SMC), perivascular cells (RGS5⁺PDGRFB⁺ pericytes, RGS5⁺PDGRFB⁻ pericytes, stimuli-responding VSMCs), and fibroblast (stressed fibroblasts and myofibroblasts) displaying the highest level of change between peri and post menopause, with more than 1,000 DE genes.

Regarding the CCC differences in perimenopause vs. postmenopause, we hypothesized that those cell types with the highest differences in signalling counts within a given network would be those that are most affected by the aging process. In brief, we discovered that stimuli-responsive SMC, canonical SMC, immune-modulated fibroblasts, and stimuli-responsive VSCM were the types most affected by the aging process, in concordance with DE results above mentioned.

This, in fact, is consistent with the overall results we present in our paper, in which we conclude that the processes most affected by aging are contractility (in terms of decreased contractility), immune responses, fibrosis, and compromised angiogenesis. This has been reflected also in the discussion session, adding a short explanation of these conclusions in page 15, lines 430-438.

However, consistent with the information presented in page 16, lines 468-473, it's important to acknowledge that while the main emphasis of this study is on the myometrium, these cell populations that seem to exhibit the most pronounced changes may also be subject to influences from other physiological processes apart from aging.

7. It is a lack of details of the computational approaches and the provided code is poorly annotated and difficult to follow on the GitHub page.

A7: The code has not been uploaded to GitHub; we apologize for the inconvenience. All the bioinformatics procedures are clearly stated in Methods section. Because no development of new algorithms, neither new implementation of pre-existent functions has been approached, code disclosure is not applicable in this work. All the R/bash/python functions applied are already published and well documented in the sources of cited packages. Disclosure of Code Availability for this study was stated in the main manuscript as Not Applicable.

Minor concerns:

1. What parameters for DoubletFinder were applied to the dataset? Were ground truths used or estimates? What pK values were used? What percentage of cells were found to be doublets?

A1: We employed estimates for DoubletFinder and determined a custom pK value for each sample by manually identifying the maxima in the mean-variance normalized bimodality coefficient plot generated using the *find.pk* function provided by the DoubletFinder package. This was added in the text in page 21, lines 633-634. We used both *DoubletFinder* and *scds* (utilizing a hybrid configuration) to identify doublets in each sample. Cells labelled as doublets by both algorithms were removed prior to sample merging. Additionally, we estimated the doublet generation rate following the 10X guidelines (Chromium Next GEM Single Cell 3' Reagent Kits v3.1), which specify that for samples with 17,000 cells loaded, the expected formation rate is 8.11%. Finally, we manually assessed the gene expression signatures of each cell cluster to eliminate clusters exhibiting a doublet-like transcriptomic profile. Figure below shows quality control statistics related to the doublet intersection ratio of the cells labelled as doublet by DoubletFinder and *scds* and the estimated number of doublets expected in each sample.

2. How were batch effects corrected? Was a design matrix added during the scaling process? Were the effects of cell cycle genes regressed out?

A2: Correction of batch effects was approached as described in Methods section “Integration of single cells and single nuclei across conditions and clustering” (page 22, line 658): “The first output of sample distribution in clusters and cluster marker genes was then explored to evaluate biases from our data batches. Next, the Harmony R package (v1.0) was used to remove four primary sources of bias: i) between single-cell and single-nuclei protocols, ii) between tissue zone of origin (anterior, posterior, or fundus), iii) between aging conditions, and iv) the patient sample origin to remove inherent inter-individual differences. Dimensions were reduced as before, and an integration diversity penalty parameter (theta) of 2 was used.”

In our data analysis workflow, the scaling process was applied with `ScaleData()` function from Seurat pipeline (see full documentation in <https://satijalab.org/seurat/>), which scales and centers features in the dataset, individually regressing out against each feature the provided variables of interest to regress. The resulting residuals are then scaled and centered. These variables to regress are given by a vector of factor/column names to the function parameter

‘vars.to.regress’. A design matrix is unnecessary here. The code used in our script for scaling and integrating (batch correction is jointly applied here) reads as follows:

```
sc_myo <- ScaleData(sc_myo, verbose = T, vars.to.regress =  
c("nFeature_RNA", "mitoRatio", "Phase"))  
  
sc_myo <- RunHarmony(sc_myo, group.by.vars = c("Protocol", "Zone",  
"Menopause_age", "Patient"), theta = c(2,2,2,2), plot_convergence = T,  
reduction.save = "harmony_Protocol_Zone_Menopause_age_Patient_theta2",  
epsilon.harmony = -Inf, assay.use = "RNA")
```

As shown in the code, during the scaling step “Phase” is included as a regressing variable, which refers to cell cycle phase. We confirm the reviewer that the effects of cell cycle genes were regressed out for all our datasets.

3. What is resolution for clustering? A statistical approach to identify the resolution or find the number of clusters should be performed. For example, a sub-sampling with bootstrapping can be used. There are a variety of packages can be used (i.e. see Patterson-Cross et al 2021 PMID: 33522897). Number of cells for each cell group and each sample should be clearly annotated.

A3: We thank the reviewer for the input on the ‘chooseR’ tool, from Patterson-Cross et al., 2021. It provides a useful advancement on the tricky task of defining the best number of clusters for a dataset. In our case we carefully used the clustree tool to generate clustering trees and evaluate the impact of varying clustering resolution and select the optimal one as a balance of resolution and noise. We have modified page 22, lines 655-656 to explain this approach in more detail. We have also added clustering resolution to figure legends of all UMAPs.

Additionally, we have added Supplementary Table 14 detailing the number of cells for each cell group and each sample.

Reviewer #3 (Remarks to the Author):

In this study, the authors present a single-cell atlas of the human myometrium in perimenopausal and postmenopausal women, illustrating the cell type-specific transcriptomic changes and alterations of cell-cell communication that accompany menopause. The authors included myometrial tissue samples from 6 perimenopausal and 14 postmenopausal women and performed both single-cell and single-nucleus RNA-sequencing. Key findings of the study include the observation of fewer contractile capillary cells and diminished expression of genes related to ion channel expression in smooth muscle cells as well as a generalized impairment of gene expression across endothelial, smooth muscle, fibroblast, perivascular, and immune cells. Furthermore, altered myometrial cell-cell communication resulting in the loss of multiple signaling pathways was a hallmark of menopause. The authors conclude that these data can provide insight into the changes underlying complications faced by older women during pregnancy and labor.

Overall, the study represents a novel application of cutting-edge single-cell techniques to an increasingly relevant topic, namely how aging can affect cellular processes in the myometrium and subsequently complicate pregnancy in older women. However, the study is highly descriptive, and the authors may consider including additional analysis to expand on the functional changes in myometrial cells.

We thank Reviewer #3 for their comments regarding the interest in our manuscript to the field and suggestions on how to improve our work. We have carefully followed your recommendations and trust that our new data and point-by-point response satisfactorily address all comments.

Specific comments are as follows:

1. A key question arising from the authors' findings is how the observed transcriptomic changes translate to functional changes in individual cell types. If possible, the authors could consider performing functional determinations in myometrial SMCs or other target cell types and comparing these between perimenopausal and postmenopausal women.

A1: We value this comment as it raises a valid point, and we fully concur that conducting functional validation of our findings would strengthen our conclusions. Therefore, after establishing the increased abundance of smooth muscle cells in the aging myometrium, we substantiated the age-related expression reduction in their contractile function using several techniques, such as spatial transcriptomics for contractility marker genes such as *SMTN* and voltage channel genes such as *KCNE4* (Fig. 4E; Supplementary Fig. 4D and E), and immunostaining (Fig. 4F) for *ACTA2/VDAC1*.

Further validations such as utilizing in vitro models, were beyond the scope of our study due to limited sample availability. However, we believe that our research lays the basis for future investigations focused on specific cell types, pathways, and genes that we have identified.

2. Regardless of the feasibility of the abovementioned functional experiments, the authors should consider further mining their generated datasets to provide more insight into the functional alterations of each cell type with menopause. For example, additional application of GO, Reactome, or other annotated pathway analysis to the differentially regulated gene sets reported for each cell type could help elucidate which functions are altered, beyond the reporting of individual genes as done by the authors. Moreover, the application of the cell-cell communication analysis could also be expanded to show the individual interactions of each cell type (e.g., SMCs) with other cells, and whether the overall interactions are altered with menopause. Such additional analyses would be complementary to the consistent structure of each results section/figure focused on specific cell types. This strategy was previously reported (PMID: 35260533).

A2: We appreciate your comment and agree that functional enrichment analysis is very valuable to evaluate which functions may be altered with aging. In fact, we used the suggested tools (GO and Reactome) to aid with the interpretation of our results and their discussion, as detailed in the Methods section (page 23, line 680). To complement and expand the results from the single-cell myometrial atlas during menopause, we have further explored cell-cell communication analysis to thoroughly investigate the extensive information obtained. With

this aim, different information outcomes in different plotting formats are displayed in Figures 1, 6, 7, and Supplementary Figure 5. To begin with, the Relative Information Flow bar plot illustrates all the pathways of cell-to-cell interaction that have shown significant changes (Figure 1D). In line with previous reporting (PMID: 35260533), we have added chord diagrams to Figures 6 and 7 to provide additional context to these findings. These diagrams allow us to visually represent the specific interactions of each cell type with other cells within the signalling pathway that exhibits significant changes between different menopausal conditions. The specific signalling pathway being depicted is identified in the title of each chord diagram. Additionally, bar plots illustrating the relative contributions of ligands and receptors within each pathway can be found in Figures 6 B, D, H, J, as well as Supplementary Figure 5. All these results have been elaborated upon and discussed throughout the manuscript.

3. It would be interesting to see how the perimenopausal and postmenopausal myometrium compare to the myometrium during “healthy” reproductive ages (<45 years or so). Given that the collection of fresh samples could take some time, would it be feasible to leverage existing datasets for comparison?

A3: Authors are thankful to the reviewer for pointing out this issue. Certainly, there is a constraint when it comes to encompassing all phases of women's reproductive lives in our study. Besides the significant time investment required to gather additional fresh samples, obtaining myometrial samples from patients without medical indications, such as uterine prolapse or from organ donor program, poses considerable challenges. Consequently, we have included an explanation in the discussion section to address this issue (page 16, lines 459-463).

We have considered the option of enhancing the study by incorporating external datasets, but there is a limited pool of available datasets suitable for comparison with our own data. As mentioned in the discussion section (page 16, lines 465-468), several publications have documented human myometrial data in distinct scenarios, including parturition (Pique-Regi et al., 2022) and leiomyoma conditions (Goad et al., 2022, GSE162122) as well as whole uteri (not focusing specifically on myometrium) at single cell level. However, these disparities in conditions render the datasets incompatible for direct comparison with ours.

4. Abstract: It is unclear why the authors specifically mention nervous system regulatory fibroblasts in the abstract, as this cell type is barely discussed in the rest of the manuscript. Consider removing to make room for more important findings.

A4: Thank you for your comment. Nervous system regulatory fibroblasts were mentioned in the abstract to highlight our efforts in defining novel cell types in the myometrium by a very careful curation of gene markers. Indeed, we have not been able to dive deeper into the role or functioning of this cell type and might seem that the focus of the paper is different if only reading the abstract. Therefore, and following reviewer 's recommendations, we have removed this from the abstract not to give this cell type more emphasis.

5. Results/Figure 1: Did the authors note underrepresentation of larger cell types (e.g., SMCs) from the single-cell dataset? Other studies of the human reproductive tissues have

noted some limitations of the microfluidics in terms of captured cell sizes (e.g., PMID: 35260533). Although the inclusion of single-nucleus sequencing would help overcome such limitations, it would still be worth mentioning any difficulties in this regard.

A5: Thank you for this comment. You are completely, right. In fact, we hypothesized that one of the reasons why we find such a “small” proportion of SMCs in a tissue that is mainly muscular could be technical limitations. In terms of smooth muscle cells (SMCs) specifically, we did notice a higher representation of this cell type in the single nuclei dataset. SMCs accounted for approximately 28% (6022 out of 21344) of single nuclei, whereas they constituted only 14% (20907 out of 139490) of single cells. We have added a paragraph regarding this limitation in page 15, lines 440-444. These cell counts are also detailed in Supplementary Table 14.

6. Results/Figure 1C: Did the authors attempt any overall comparative analysis between spatial transcriptomic maps of the myometrium? It would be interesting to note if trends for differing localization of major cell clusters could be observed post-menopause, as this could partly relate to the reported changes in cell-cell communication.

A6: Thank you for this comment. Certainly, we conducted a comparison of cell population distributions between peri- and post-menopausal samples. These findings will be elaborated upon in subsequent sections, with a particular focus on cell subpopulations. Additionally, we share the interest in exploring potential differences in cluster localization with aging. However, it's worth noting that spatial transcriptomics is a relatively new field, and specific analysis tools for comparing conditions are currently under development.

Nevertheless, in an attempt to address the requirements suggested by the reviewer, we have assessed whether there are changes in the distribution of the main cell populations based on their proximity to the endometrium. We have referred to these areas as the "distal" and "proximal" regions. The most noteworthy outcome of this innovative approach was that the variation in smooth muscle cell (SMC) proportion between peri and postmenopause primarily occurred in the "distal" region, with no discernible difference in the "proximal" region. These novel findings have been included in the manuscript with the following modifications:

- A comprehensive explanation of this analysis approach has provided in the Methods section (page 24, lines 718-725).
- A new boxplot comparing tissue regions and peri or postmenopausal status has been incorporated into Supplementary Figure 4 and mentioned in the Results section (page 7, lines 189-191).
- These findings have been also commented in the Discussion section by 1) explaining the possible causes of these differences in regions in page 12, line 356-360 and 2) by giving a possible reasoning for the inconsistency between scRNAseq and spatial transcriptomics in (page 15, lines 449-457).

Additionally, we have integrated spatial transcriptomics data with CCC data from the scRNAseq analysis, which have also been incorporated to the manuscript (page 10, lines 269-272).

7. Results/Figure 1D: The term “information flow” should be explained in the results section, as this may not be immediately apparent to readers.

A7: Thank you for your comment. We have added a small clarification in page 5, lines 120-122 to improve text comprehension.

8. Results/Supplementary Figure 3: The spatial transcriptomics should include quantification, as shown in other figures, to support the specific cell types or genes mentioned.

A8: We appreciate your feedback, and we agree with your suggestion. We have included quantification boxplots in Supplementary Figure 4 (formerly Supplementary Figure 3) to provide additional support for our observations concerning SMC proportion and SMTN expression. Additionally, in response to your prior input, we have adopted a more spatially-oriented approach, specifically comparing the distal and proximal myometrium in relation to the endometrium.

9. Results/Figure 5: What are the characteristics of the “undifferentiated” T cells? Is it possible that this subset includes innate lymphoid cells or MAIT cells?

A9: Thank you for your comment. After reviewing the text, we have realized that this was in fact a mistake and we have rectified it in the text (line 222), since we should refer to “T-cells”. This is because this specific cluster exhibited the transcriptomic profile and gene markers of a T-cell. We did, in fact, conduct a search for MAIT cells; however, none of the identified cell clusters displayed the canonical markers associated with these cells such as TRAV1-2, CD161 (Godfrey et al. 2019). Moreover, due to insufficient information, we were unable to classify T cells as CD4+, CD8+, or undifferentiated. As a result, we opted for a broader label.

10. Figure 1: For Figure 1B, please move the color legends to be close to the top of the figure panel, to aid in interpreting each color for the pie charts and cell types. In addition, there is unnecessary white space between Figure 1B and 1D that could be reduced to improve the “squareness” of the figure.

A10: As suggested, we have relocated the color legends near to the upper section of the figure panel, to aid in interpreting each color for the pie charts and cell types. We have also minimized the space between Figure 1B and 1D to improve the “squareness” of the figure.

11. Figure 2: For the plot types used in Figure 2B-C (similar plots are in other figures as well) it is not immediately apparent how the colors and changes relate to the study groups (i.e., what is increased/decreased with menopause). Some additional labelling or other modifications would be helpful. In addition, the cluster names in Figure 2D should be larger, as these are currently too similar to other less important plot labels.

A11: To facilitate the interpretation of cell abundance analysis, we have added a new color-coded scale bar for Figure 2B-C, as well as Figure 4B-C, which corresponds to different aging conditions.

12. Line 288 has a typo, the figure citation here is likely meant to be Fig. 7E.

A12: Thank you for pointing this out. We have corrected the figure citation to Fig. 7E in Line 313.

THUMBNAIL LIST OF FIGURES/ SUPPLEMENTARY FIGURES

SUPPLEMENTARY FIGURES		
REVISED MANUSCRIPT	FORMER MANUSCRIPT	CONTENT
Fig.1	Fig.1	Original Fig.1 after moving the color legends to the top of the figure panel 1B and reducing the space between Figure 1B and 1D to improve the “squareness” of the figure.
Fig.2	Fig.2	Original Fig.2 including new scale bar of colour codes associated to aging conditions.
Fig.4	Fig.4	Original Fig.4 including new scale bar of colour codes associated to aging conditions.
Supplementary Fig.1	Supplementary Fig.1	Original Supplementary Fig.1 incorporating cell distribution from the different uterine areas (fundus and anterior and posterior)
Supplementary Fig.2 (new)		UMAP and barplots depicting cell distribution based on clinical variables
Supplementary Fig.3	Supplementary Fig.2	Single-cell expression and Spatial location
Supplementary Fig.4	Supplementary Fig.3	Original Supplementary Fig.3 incorporating quantification boxplots of spatial data and differences in SMC proportion between the distal

		and proximal regions of the myometrium
Supplementary Fig.5	Supplementary Fig.4	Differential age-related changes in cell-to-cell communication
Supplementary Fig.6 (new)		Integration of CCC and spatial transcriptomics data
Supplementary Fig.7 (new)		UMAP, ST and IHQ showing the expression of ESR1/PGR in human myometrium

LIST OF TABLES/ SUPPLEMENTARY TABLES

SUPPLEMENTARY TABLES		
REVISED MANUSCRIPT	FORMER MANUSCRIPT	CONTENT
Supplementary Table 1 (new)	Supplementary Table 1	Clinical metadata
Supplementary Table 14 (new)		Number of cells per group, sample and cell type

REFERENCES

Anamthathmakula P, Kyathanahalli C, Ingles J, Hassan SS, Condon JC, Jeyasuria P. Estrogen receptor alpha isoform ERdelta7 in myometrium modulates uterine quiescence during pregnancy. *EBioMedicine*. 2019 Jan; 39:520-530. doi: 10.1016/j.ebiom.2018.11.038. Epub 2018 Nov 28. PMID: 30502052; PMCID: PMC6355643.

Bhartiya D. Adult tissue-resident stem cells-fact or fiction? *Stem Cell Res Ther*. 2021 Jan 21;12(1):73. doi: 10.1186/s13287-021-02142-x. PMID: 33478531; PMCID: PMC7819245.

Bornstein, E., Eliner, Y., Chervenak, F. A. & Grünebaum, A. Concerning trends in maternal risk factors in the United States: 1989–2018. *EClinicalMedicine* 29–30, (2020).

Cavalcante, M. B., Saccon, T. D., Nunes, A. D. C., Kirkland, J. L., Tchkonja, T., Scheider, A., & Masternak, M. M. (2020). Dasatinib plus quercetin prevents uterine age-related dysfunction and fibrosis in mice. *Aging*, 12(3), 2711–2722.

Elmes, M. et al. Maternal age effects on myometrial expression of contractile proteins, uterine gene expression, and contractile activity during labor in the rat. *Physiol Rep* 3, (2015).

Davezac M, Buscato M, Zahreddine R, Lacolley P, Henrion D, Lenfant F, Arnal JF, Fontaine C. Estrogen Receptor and Vascular Aging. *Front Aging*. 2021 Sep 24;2:727380. doi: 10.3389/fragi.2021.727380. PMID: 35821994; PMCID: PMC9261451.

Fafián-Labora JA, O'Loughlen A. Classical and Nonclassical Intercellular Communication in Senescence and Ageing. *Trends Cell Biol*. 2020 Aug;30(8):628-639. doi: 10.1016/j.tcb.2020.05.003. Epub 2020 Jun 3. PMID: 32505550.

Goad J, Rudolph J, Zandigohar M, Tae M, Dai Y, Wei JJ, Bulun SE, Chakravarti D, Rajkovic A. Single-cell sequencing reveals novel cellular heterogeneity in uterine leiomyomas. *Hum*

Reprod. 2022 Sep 30;37(10):2334-2349. doi: 10.1093/humrep/deac183. PMID: 36001050; PMCID: PMC9802286.

Godfrey DI, Koay HF, McCluskey J, Gherardin NA. The biology and functional importance of MAIT cells. *Nat Immunol*. 2019 Sep;20(9):1110-1128. doi: 10.1038/s41590-019-0444-8. Epub 2019 Aug 12. PMID: 31406380.

Hessler, S. C. et al. Myometrial artery calcifications and aging. *Menopause*. 2015. Dec;22, 1285–1288. doi: 10.1097/GME.0000000000000475

Kissler, K. & Hurt, K. J. The Pathophysiology of Labor Dystocia: Theme with Variations. *Reproductive sciences* (2022) doi:10.1007/s43032-022-01018-6.

López-Otín C, Blasco MA, Partridge L, Serrano M, Kroemer G. Hallmarks of aging: An expanding universe. *Cell*. 2023 Jan 19;186(2):243-278. doi: 10.1016/j.cell.2022.11.001. Epub 2023 Jan 3. PMID: 36599349.

Ma Y, Zhou X. Spatially informed cell-type deconvolution for spatial transcriptomics. *Nat Biotechnol*. 2022 Sep;40(9):1349-1359. doi: 10.1038/s41587-022-01273-7. Epub 2022 May 2. PMID: 35501392; PMCID: PMC9464662.

Marquez, C. M. D., Ibane, J. A. & Velarde, M. C. The female reproduction and senescence nexus. *American Journal of Reproductive Immunology* 77, (2017).

Mas A, Nair S, Laknaur A, Simón C, Diamond MP, Al-Hendy A. Stro-1/CD44 as putative human myometrial and fibroid stem cell markers. *Fertil Steril*. 2015;104(1):225-34.e3.

Ono M, Maruyama T, Masuda H, Kajitani T, Nagashima T, Arase T, Ito M, Ohta K, Uchida H, Asada H, Yoshimura Y, Okano H, Matsuzaki Y. Side population in human uterine myometrium displays phenotypic and functional characteristics of myometrial stem cells. *Proc Natl Acad Sci U S A*. 2007 Nov 20;104(47):18700-5. doi: 10.1073/pnas.0704472104.

Ono M, Kajitani T, Uchida H, Arase T, Oda H, Uchida S, Ota K, Nagashima T, Masuda H, Miyazaki K, Asada H, Hida N, Mabuchi Y, Morikawa S, Ito M, Bulun SE, Okano H, Matsuzaki Y, Yoshimura Y, Maruyama T. CD34 and CD49f Double-Positive and Lineage Marker-Negative Cells Isolated from Human Myometrium Exhibit Stem Cell-Like Properties Involved in Pregnancy-Induced Uterine Remodeling. *Biol Reprod*. 2015 Aug;93(2):37. doi: 10.1095/biolreprod.114.127126.

Paul EN, Carpenter TJ, Fitch S, Sheridan R, Lau KH, Arora R, Teixeira JM. Cysteine-Rich Intestinal Protein 1 is a Novel Surface Marker for Myometrial Stem/Progenitor Cells. *bioRxiv* [Preprint]. 2023 Mar 18:2023.02.20.529273. doi: 10.1101/2023.02.20.529273.

Peavey MC, Wu SP, Li R, Liu J, Emery OM, Wang T, Zhou L, Wetendorf M, Yallampalli C, Gibbons WE, Lydon JP, DeMayo FJ. Progesterone receptor isoform B regulates the Oxtr-Plcl2-Trpc3 pathway to suppress uterine contractility. *Proc Natl Acad Sci U S A*. 2021 Mar 16;118(11):e2011643118. doi: 10.1073/pnas.2011643118. PMID: 33707208; PMCID: PMC7980420.

Rosenthal, A. N., & Peterson-Brown, S. (1998). Is there an incremental rise in the risk of obstetric intervention with increasing maternal age? *British Journal of Obstetrics and Gynaecology*, 105, 1064–1609.

Sheen, J. J. et al. Maternal age and risk for adverse outcomes. *Am J Obstet Gynecol* 219, 390.e1-390.e15 (2018).

Simmen, R. C. M. et al. The krüppel-like factors in female reproductive system pathologies. *J Mol Endocrinol* 54, R89–R101 (2015).

Spies D, Renz PF, Beyer TA, Ciaudo C. Comparative analysis of differential gene expression tools for RNA sequencing time course data. *Brief Bioinform*. 2019 Jan 18;20(1):288-298. doi: 10.1093/bib/bbx115. PMID: 29028903; PMCID: PMC6357553.

Tabula Sapiens Consortium*The Tabula Sapiens: A multiple-organ, single-cell transcriptomic atlas of humans. *Science*. 2022 May 13;376(6594):eabl4896. doi: 10.1126/science.abl4896. Epub 2022 May 13. PMID: 35549404; PMCID: PMC9812260.

REVIEWER COMMENTS

Reviewer #1 (Remarks to the Author):

The authors have well responded to the reviewers' comments. The manuscript is significantly improved.

Reviewer #2 (Remarks to the Author):

In the revised manuscript, it is evident that the authors have made some modifications in response to the previous concerns. However, it is disappointing to note that the revisions do not effectively address the issues raised by reviewers and the overall improvement of the manuscript is limited.

The proposed conclusion still lacks the necessary experimental validation or comparative analysis. For example, the factors responsible for myometrium aging and their role as biomarkers or drivers remain unclear without independent experimental validation. Additionally, there is limited analysis of spatial transcriptome analysis in the current manuscript.

Thus, the revisions presented do not effectively address the raised concerns. The overall quality of the paper has not been significantly improved to meet the high standards of a decent journal like Nature Communications, and additional substantial revisions are necessary.

Reviewer #3 (Remarks to the Author):

Thank you for addressing my comments and concerns.

We would like to express our gratitude again to the three reviewers for assessing our revised manuscript. Reviewers #1 and #3 indicated that they satisfied with the revised version of the manuscript.

Please find below our response to Reviewer #2's comments:

R1. In the revised manuscript, it is evident that the authors have made some modifications in response to the previous concerns. However, it is disappointing to note that the revisions do not effectively address the issues raised by reviewers and the overall improvement of the manuscript is limited.

A1. Reviewer #2 has made several valuable comments that have led to additional experimentation, analysis, and this information has been incorporated in the previous revision of the manuscript.

In a nutshell, please note that we have **(1)** substantiated the age-related expression reduction in their contractile function using spatial transcriptomics for contractility marker genes such as *SMTN* and voltage channel genes such as *KCNE4*, and immunostaining for *ACTA2*/*VDAC1* generating new figures (Fig. 4E & 4F; Supplementary Fig. 4D and E) and Supplementary Table 8, **(2)** performed analyses of estrogen and progesterone receptors mRNA and protein (by immunostaining shown in New Supplementary Figure 7), **(3)** added an integrative analysis of single-cell RNA seq and spatial transcriptomics with tissue validation (New Supplementary Figure 6), and **(4)** compared our results to those of other published single-cell atlases of human myometrium (page 12, line 331).

To facilitate your evaluation of our efforts, please see below a detailed table with the reviewer requests and our itemized reply with the actions taken:

Original request	Analysis/experiment conducted	Results summary	New data provided
R1: Young patient samples should be included to better understand the age-molecular basis in human myometrium.	Since it is not feasible to expand the study including younger individuals (previously argued), evaluation of publicly available datasets (myometrium single-cell atlas) was performed to bridge the data gap.	Our data is not comparable with other myometrial atlas, since they are focused on myometria with fibroids (Goad et al., 2022); pregnant myometria (Regi et al., 2022) and the uterus as a whole (Tabula Sapiens, 2022) with very few young patients in each of them.	Included in the discussion section (page 16, lines 465-468).
R2: Clarify detailed information for each donor and analyse how age-related menstrual cycle dysregulation contributes to human myometrium aging.	i) We have thoroughly investigated the clinical data of the patients. ii) Analysing the impact of menstrual phases poses difficulties due to hormonal fluctuations and the unpredictability of menstrual cycles in peri-menopausal patients, as well as limited access to information from organ donors. However, we have explored the expression of oestrogen and progesterone receptors (ESR1, ESR2 and PGR) at mRNA (scRNA-seq and spatial transcriptomics) and protein level (immunostaining).	ii) While ESR2 did not exhibit any differential expression, our findings revealed an upregulation of ESR1 in all cell subpopulations during post-menopause, as evidenced by our results from both single-cell transcriptomics and spatial analysis. Conversely, PGR displayed a downregulation specifically in contractile smooth muscle cells (SMC) during post-menopause, as indicated by the results of the differential expression analysis, which was further supported by spatial representations.	i) We have provided a detailed and update Table 1. ii) We have created a new Supp. Fig 7. Supporting findings have been also presented in the discussion section (page 14 lines 419-429).
R3: In-depth analysis into the bioinformatic data and mine the core regulatory mechanism underlying human myometrium aging, including experimental validations.	i) Differential abundance and differential expression analysis have been performed. ii) We substantiated the age-related expression reduction in their contractile function using spatial transcriptomics for contractility marker genes such as SMTN and voltage channel genes such as KCNE4, and immunostaining for ACTA2/ VDACL1.	i) We have showed that uterine zones in term of cell abundance were uniformly distributed with no significant differences. ii) Comparative analyses revealed a significant reduced expression of contractility-associated genes in all SMCs during myometrial aging. Interestingly, we found a significant downregulation in K+ voltage channel gene expression in the postmenopausal myometrium.	i) We have created a new Supp. Fig 1C. ii) We have generated new figures (Fig. 4E & 4F; Supplementary Fig. 4D and E and Supplementary Table 8).

R4: Investigate the spatial-specific changes during the aging process.	We have performed: i) Comparative analysis of cell proportions and gene expression between peri/post-menopausal samples. ii) Comparative analysis of cell proportions between the distal and proximal regions of the myometrium, taking the endometrium as a reference point.	Spatial transcriptomics analysis revealed a change in gene expression and SMC proportion between peri and post-menopause that was only appreciable in the distal region of the myometrium from the endometrium. These findings are in line with previous reports highlighting the relevance of the endometrial-miometrial junctional zone in reproductive outcomes and changes in this region that take place with the aging process.	The new information has shown in the methods section (page 24, lines 718-725). New figures have been included (Fig 2E-F; Fig 3D-E; Fig 4E; Fig 5C) Moreover, new images and boxplot comparing tissue regions has been included in Supp. Fig 4B-D and Supp. Fig 7C-F.
R5: Integrative analysis of single-cell transcriptome and spatial transcriptome data.	We performed spatial-informed cell-to-cell communication analysis of the main cell types presents in the tissue; spatial transcriptomics data was implemented using the CellChat library.	Validation of CCC in several signaling networks (ANGPTL, CXCL, and PDGF) among major cell types (SMC, fibroblasts, VSMC, and endothelium) at tissue level, locating the direction and strength of each interaction.	The new information has been depicted in the methods (page 25, lines 740-745). Supp. Fig 6 has been also added.
R6: Which cells are the most sensitive to aging and likely to contribute to tissue aging.	We have performed in depth-review of differential expression and cell to cell communication results by comparison of the number of differentially expressed genes and numbers of interactions.	We have shown that SMCs, perivascular cells and fibroblasts are the most affected by the aging process. It aligns with the findings of reduced contractility, angiogenesis, and increased fibrosis, as further discussed in our paper.	Included in the discussion section (page 15, lines 430-438).

Minor concerns regarding the bioinformatic analysis.	We have provided further details on the bioinformatic analysis:  i) DoubletFinder parameters, pK values and percentage of doublets. ii) Batch correction, design matrix and regression of cell cycle genes. iii) Clustering resolution and number of cells per group and cell type. 	 i) pK values were estimated using the find.pk function and provide quality control statistics related to the doubled intersection ratio of doublets and the number of estimated doublets. ii) Harmony was used for data integration and batch correction between protocols (single-cell and single-nuclei); tissue zone (anterior, posterior or fundus); aging condition (peri or post menopause) and patient sample origin. The data was also scaled and centered, including cell cycle phase as a regressing variable. iii) We used the clustree tool to generate clustering trees and evaluate the impact of varying clustering resolution and select the optimal one as a balance of resolution and noise. 	 i) Described in (page 21, lines 633-634) and specific figure related to the analysis was provided in the previous response version. ii) Described in Methods section (page 22, line 658). iii) We have modified page 22, lines 655-656 to explain this approach in more detail. We have also added clustering resolution to figure legends of all UMAPs. Supp. Table 14 has been also added.
---	--	--	---

R2. The proposed conclusion still lacks the necessary experimental validation or comparative analysis. For example, the factors responsible for myometrium aging and their role as biomarkers or drivers remain unclear without independent experimental validation. Additionally, there is limited analysis of spatial transcriptome analysis in the current manuscript.

A2. The additional analyses in response to the comments of Reviewer #2 has led us to identify subsets of smooth muscle cells (stimuli-responsive and canonical) as well as fibroblasts and vascular smooth muscle cells as potentially important in the impairment of myometrial contractility, immune response, fibrosis, and angiogenesis with aging. These findings are all novel and conclude that the processes most impacted by aging in the myometrium include decreased contractility, altered immune responses, increased fibrosis, and impaired angiogenesis (page 15, line 436).

Now, reviewer #2 would like to have independent validation of our findings. Obtaining samples of myometrium from young women is an extremely difficult challenge because the circumstances that would allow collection of these samples (hysterectomies or deceased donors) are rare. Indeed, the samples that we presented as part of the study took three years to assemble. The Reviewer also asked for functional studies. The key function of the myometrium is contractility. Functional studies of myometrium require fresh tissue and, therefore, this would represent an extraordinary undertaking well beyond the scope of what we had originally planned. Moreover, by addressing the reviewer #2 concern regarding the spatial transcriptome analysis, we have exerted special efforts to maximize the scientific value from the results obtained, given the current limitations of publicly available software and existing bioinformatics tools in this emerging field.

While acknowledging the aforementioned limitations, we have restructured the final paragraph of our manuscript to underscore the substantial enhancement resulting from the incorporation of the reviewer's comments. We have conducted extensive functional and in-silico validations to the furthest of our possibilities, providing a comprehensive description of the molecular and cellular profile associated with the age-related myometrial dysfunction (page 3, line 74).

R3. Thus, the revisions presented do not effectively address the raised concerns. The overall quality of the paper has not been significantly improved to meet the high standards of a decent journal like Nature Communications, and additional substantial revisions are necessary.

A3. The impetus for our study was to understand at the single-cell level why older women have more cesarean deliveries, obstetrical complications such as postpartum hemorrhage leading to 7,8-fold-increase in maternal mortality compared to women under 40 (Hoyert, 2019; Bornstein et al., 2020). The pathways identified in our study are remarkably similar to those reported to be key for parturition by the group at NIH when studying myometrium in parturition at single-cell resolution (Pique-Regi et al., JCI Insight 7, e153921 (2022)).

Despite the limitations indicated above, we consider that our approach remains sufficiently robust, and the results provide novel insights for a better understanding of the mechanisms underlying aged myometrium and its impact on obstetric complications as an important burden in women's health.

REVIEWERS' COMMENTS

Reviewer #2 (Remarks to the Author):

Despite the authors' extensive in-silico analysis and validations, the proposed conclusions lack the crucial support of experimental verification or comparative analysis to uncover the underlying drivers or factors contributing to myometrium aging. This gap in the research inhibits a comprehensive understanding of the mechanisms involved. To advance this field, it is imperative to conduct further experiments that can validate and complement the computational findings.

Moreover, the identification of biomarkers that can accurately indicate myometrium aging and their significance in terms of diagnosis and prevention requires additional investigation. Without robust experimental evidence linking these biomarkers to myometrium aging, their clinical relevance and potential applications remain uncertain. Additionally, the limited integration of spatial transcriptome and sc/snRNA-seq data represents a missed opportunity to fully exploit the advantages offered by spatial transcriptome analysis. By neglecting the potential synergies between these two techniques, the study fails to maximize the wealth of information that could be gained from a comprehensive approach.

In summary, while the authors have made commendable efforts in their in-silico analysis, their work would benefit from a stronger foundation provided by detailed experimental verification, comparative analysis, and the integration of spatial transcriptome and sc/snRNA-seq data. Addressing these limitations will undoubtedly enhance the scientific rigor, significance, and translational potential of the study.